# BNMusic: Blending Environmental Noises into Personalized Music

**Chi Zuo**[1], **Martin B. Møller**[2], **Pablo Martínez-Nuevo**[2], **Huayang Huang**[1], **Yu Wu**[*,1], **Ye Zhu**[3,4]

[1]School of Computer Science, Wuhan University, China
[2]Bang & Olufsen A/S, Denmark
[3]Department of Computer Science, Princeton University, USA
[4]LIX, École Polytechnique, IP Paris, France
{zuoc97,wuyucs,hyhuang}@whu.edu.cn
{mim,pmn}@bang-olufsen.dk
ye.zhu@polytechnique.edu

## Abstract

While being disturbed by environmental noises, the acoustic masking technique is a conventional way to reduce the annoyance in audio engineering that seeks to cover up the noises with other dominant yet less intrusive sounds. However, misalignment between the dominant sound and the noise—such as mismatched downbeats—often requires an excessive volume increase to achieve effective masking. Motivated by recent advances in cross-modal generation, in this work, we introduce an alternative method to acoustic masking, aiming to reduce the noticeability of environmental noises by blending them into personalized music generated based on user-provided text prompts. Following the paradigm of music generation using mel-spectrogram representations, we propose a ***Blending Noises into Personalized Music (BNMusic)*** framework with two key stages. The first stage synthesizes a complete piece of music in a mel-spectrogram representation that encapsulates the musical essence of the noise. In the second stage, we adaptively amplify the generated music segment to further reduce noise perception and enhance the blending effectiveness, while preserving auditory quality. Our experiments with comprehensive evaluations on MusicBench, EPIC-SOUNDS, and ESC-50 demonstrate the effectiveness of our framework, highlighting the ability to blend environmental noise with rhythmically aligned, adaptively amplified, and enjoyable music segments, minimizing the noticeability of the noise, thereby improving overall acoustic experiences. Project page: https://d-fas.github.io/BNMusic_page/.

## 1   Introduction

In public environments like subway trains, passengers are often exposed to persistent and irritating noise. While active noise cancellation (ANC) [12] is effective in personal audio devices, its individual-oriented nature limits its practicality in group settings. Equipping every passenger with ANC headphones is unrealistic, and such systems often struggle with high-frequency noise. To address this, we propose a new task: rather than eliminating noise for individuals through destructive interference techniques, we aim to blend the environmental noise with correctly designed music in a way that reduces its perceptual salience for a group of listeners, as shown in Fig.1 (a). This perceptual blending shifts the goal from directly suppressing noise to reducing its impact through harmonious audio masking, enabling a scalable auditory enhancement in shared environments, providing a more comfortable auditory experience without requiring personal devices. Beyond public transportation,

---

*Corresponding author.

39th Conference on Neural Information Processing Systems (NeurIPS 2025).

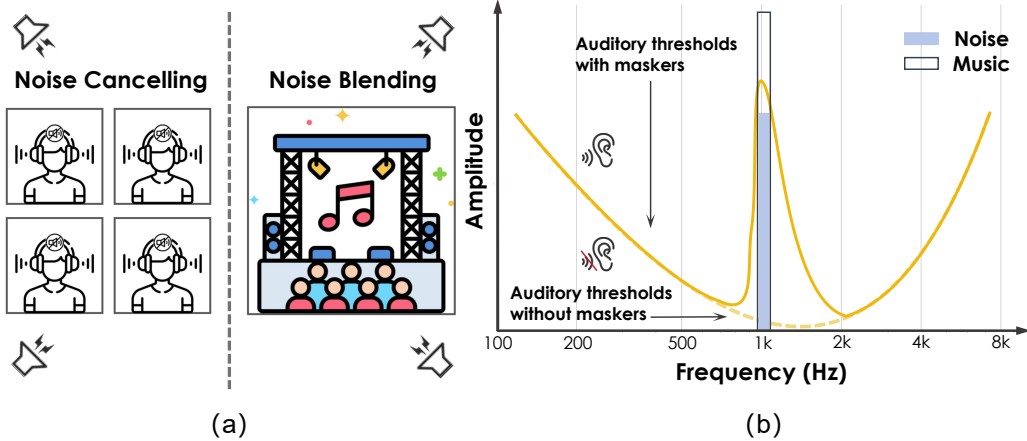

(a)                                                                 (b)

Figure 1: **(a):** Noise cancelling is designed for individual use, requiring proximity to the user, while our **noise blending** aims to reduce noise annoyance for everyone in the room by seamlessly blending a complementary sound with the surrounding noise. **(b):** This figure demonstrates the principle of **auditory masking** in psychoacoustics [28]. The yellow dashed line indicates the threshold in quiet, i.e., the baseline level below which sounds are inaudible in the absence of other stimuli. When a music signal is introduced, it elevates the masking threshold in a frequency-dependent manner, shown by the yellow solid line. As a result, any concurrent noise components that fall below this elevated threshold become *imperceptible* to the listener, effectively masked by the presence of the music.

this blending-based approach can also be valuable in a variety of other noise-prone environments. For instance, elevators often produce repetitive mechanical sounds during operation. Similarly, household appliances such as washing machines or automatic garage doors generate rhythmic, ongoing noise. In such settings, masking the noise with well-aligned music would help improve user comfort and overall auditory experience.

Our proposed task draws inspiration from the theory of auditory masking [7, 34, 28], a psychoacoustic phenomenon where the perception of one sound is reduced or eliminated by the presence of another sound. This effect is typically modeled in the frequency domain using psychoacoustic principles, which define a masking threshold—the minimum intensity below which a sound becomes inaudible to the human ear in the presence of a masker [5]. When a new sound is introduced, it elevates the masking threshold not only at its own frequency but also across neighboring frequency bands, making weaker signals in those regions imperceptible [21]. As shown in Fig. 1 (b), such thresholds are fundamental to many perceptual audio models, and here we use them to guide the blending of generated music with environmental noise. This task aims to **diminish the perception of unwanted noise by introducing supplementary background musical sounds**. In practice, fully masking complex environmental noise within a comfortable loudness range is rarely achievable. Instead, we aim for a more feasible and perceptually effective solution through blending. By generating music that is rhythmically aligned with the underlying noise, our approach enables partial masking while incorporating the residual noise components into the musical texture, thus reducing perceived annoyance without overwhelming the listener.

To effectively blend generated music with environmental noise, it is essential that the music can perceptually mask the noise, thereby reducing its distracting impact. Achieving such blending requires that the masking effect remain strong even at relatively low overall loudness levels. To this end, the generated music should align closely with the noise in terms of rhythm or other structural properties, enabling it to integrate naturally without relying on excessive loudness. However, while recent advances in music generation, particularly those based on mel-spectrogram representations in the frequency domain [6, 35, 19, 14, 39, 40, 43, 17], have demonstrated remarkable progress, most existing models are trained to generate music from clean, structured inputs such as text prompts or orderly music excerpts. These models *struggle when conditioned on noisy and unstructured inputs*, as they lack the ability to retain relevant acoustic attributes from chaotic signals. Yet they also provide a natural foundation for leveraging masking principles, which are inherently frequency-dependent. Building on this emerging paradigm, we explore generating music in the frequency domain whose

structure, rhythmic patterns, and statistical properties naturally align with those of the noise, enabling seamless auditory blending, thus in turn diminishes the listeners' awareness of the underlying noise.

In this paper, we propose a novel approach ***Blending Noises into Personalized Music (BNMusic)*** that uses adaptive loudness-amplified music to blend with background noise. This approach makes the noise less noticeable while maintaining a balanced overall volume, thereby enhancing the acoustic environment effectively. Our method is designed as a two-stage process that explicitly targets the high-energy regions of noise, which are the most perceptually salient and thus the hardest to mask. In the first stage, we condition on these regions and apply a two-step outpainting–inpainting process on the noise mel-spectrogram to generate music that rhythmically and spectrally aligns with the noise. Outpainting extends musical patterns around the dominant noise zones, while inpainting reconstructs coherent content within them, ensuring that the resulting composition inherits the noise's most prominent frequency structures in a natural way. This design effectively transforms disruptive noise components into elements that can be musically integrated. In the second stage, we apply adaptive amplification targeted to these same regions. Because the generated music already shares temporal–spectral characteristics with the noise, only modest gain is required to strengthen the masking effect, avoiding excessive loudness while maintaining musical balance. Together, the two stages form a tightly coupled system: Stage 1 embeds the noise into the music in a perceptually aligned manner, and Stage 2 consolidates this alignment by enhancing masking where it matters most. As a result, BNMusic achieves robust blending that suppresses the perceptual salience of noise while preserving coherence and listening comfort.

To assess the efficacy of our approach in blending with environmental noise and creating a more pleasant auditory experience, we conduct comprehensive objective and subjective evaluations. We use EPIC-SOUNDS[10] and ESC-50[31] as noise sources, which together cover a wide range of real-world environmental sounds, including various actions, objects, and acoustic scenes, spanning a broad frequency range from 100 Hz to 10,000 Hz. Results show that our complete method achieves the best performance on MusicBench [19].

In summary, we present three major contributions:

- We introduce a novel multimodal generation task, namely the *noise blending with music*, whose primary objective is to reduce the perceptibility of environmental noise with personalized generated music compositions based on user-providential prompts.
- We propose **BNMusic** to construct music that integrates musical elements from the noise using a two-step outpainting and inpainting process and then utilize the auditory masking effect to adaptively amplify the generated music segment to blend with ambient noise and minimize its noticeability, creating a more pleasant acoustic environment.
- Extensive experiments demonstrated our method effectively generates music that seamlessly blends with environmental noise, minimizing its perception at applicable volume levels.

## 2   Related work

### 2.1   Conventional acoustic methods

The conventional acoustic methods of enhancing hearing environments have evolved significantly throughout these years, driven by the demand for better auditory experiences. Early efforts focused on employing physical barriers and materials to block out noise. The introduction of active noise canceling (ANC) [12] marked a major breakthrough, which utilizes microphones and electronic circuitry to produce anti-phase sound waves that counteract noise [36]. Researchers have also studied auditory masking for years [7, 34], which describes how a louder sound can reduce or eliminate the perception of a quieter one occurring simultaneously. Building psychoacoustic models to simulate the human perception of audio [28, 24, 22] and quantifying how much louder a masker needs to be at different frequencies to effectively mask a target sound. Studies have applied auditory masking principles to reduce the perceptual impact of noise, designing comfortable soundscapes using chord progressions and melodies that align with the peak frequencies of disruptive noises [23] and investigating the use of natural sounds to mask traffic noise in urban environments [42]. Or utilizing audio masking for audio watermarking [38]. Inspired by these masking-based techniques, we propose the noise blending task that uses the auditory masking effect to reduce the perceptual impact of noise on acoustic environments.

## 2.2 Generative models for music

Generative models have recently achieved remarkable success in both vision and audio synthesis [6, 13, 35, 19, 29, 4, 44]. In image generation, diffusion models [9, 32] have proven especially effective by gradually denoising in the latent space of a pre-trained network, leveraging its structure to produce high-quality outputs. Inspired by this success, diffusion-based techniques have also been extended to Text-to-Music tasks [13, 35, 19, 44], significantly improving both generation quality and efficiency. Contemporary audio generation models generally fall into two categories: those that generate audio directly in the waveform domain, and those that first generate time-frequency representations (e.g., mel-spectrograms) and convert them to waveforms using a vocoder. MusicGen [3] exemplifies the former, synthesizing music through multiple streams of discrete audio tokens. In contrast, Riffusion [6] follows the latter approach, generating mel-spectrograms via a fine-tuned Stable Diffusion [32] model and converting them into audio. AudioLDM [13] builds on latent-space modeling and CLAP (Contrastive Language-Audio Pretraining) to generate audio from text. Mustango [19] introduces MuNet for fine-grained control, achieving competitive performance even with limited training data. AudioLDM2 [14] proposes a unified "language of audio" (LOA) representation, using GPT-2 to bridge multiple modalities and guiding generation with a latent diffusion model. Our work builds on these foundations, aiming to generate music that blends harmoniously with environmental noise using diffusion-based methods. Meanwhile, complementary efforts have focused on accelerating generation for real-time applications [26, 33, 25, 2], which may enhance the practical deployment of our method in future iterations.

## 2.3 Generation controlling techniques

Recent years have witnessed significant progress in audio editing, with a growing number of works exploring the potential of generative models in transforming and manipulating sound [39, 20, 1, 18, 40, 43, 17]. Many approaches adopt a vision-inspired paradigm by converting audio into 2D representations such as mel-spectrograms, enabling the application of powerful image editing techniques to audio [6, 35, 19, 14, 39, 40, 43, 17]. Following the advances in image generative models like Stable Diffusion [32], audio editing has expanded to include tasks such as spectrogram inpainting, style transfer, and attribute control. For instance, Audit [39] applied latent diffusion models (LDMs) to edit audio in a controllable manner. Prior audio inpainting methods [20, 1, 18] typically focus on interpolating the gap between two audio clips. In contrast, more recent works [17, 43, 8, 15] leverage spectrogram-based image diffusion to manipulate high-level features such as genre, instrument, or mood. Inspired by inpainting tasks in vision and their success in image composition and context-aware filling [16, 41], we propose a new application scenario in which spectrogram inpainting and outpainting are used to mask environmental noise through musical blending. Built upon Riffusion [6], our method adapts image-style spectrogram generation to synthesize rhythmically aligned, stylistically coherent music that perceptually reduces the annoyance of background noise. Unlike traditional gap-filling methods, our approach treats the noise-occupied spectrogram as a canvas to be expanded and enhanced, thus offering a novel direction in content-aware audio editing.

## 3  *BNMusic* framework: Blending Noises into personalized Music

In this section, we present the details of our *BNMusic* framework, designed to blend noise $A_{\text{Noise}}$ with adaptive amplified music $A_{\text{Music}}$ generated from $A_{\text{Noise}}$ and text condition $C_{\text{text}}$. Our method extends the existing model's application without additional training.

**Problem statement.**  We formalize the noise blending task as an alternative to traditional masking methods, which often require excessive volume to reduce the annoyance of background noise. Given a repeating noise segment $A_{\text{Noise}}$, our goal is to generate a music segment $A_{\text{Music}}$ conditioned on both $A_{\text{Noise}}$ and a user-provided prompt $C_{\text{text}}$. When played alongside the ambient noise, $A_{\text{Music}}$ is expected to effectively reduce the noticeability of some major parts of the noise, making the remaining noticeable content less irritating or even being recognized as part of the music, and consequently, enhancing the overall auditory experience. To achieve this, we propose the *BNMusic* framework. As shown in Fig. 2, the masked noise reveals a regular rhythm or pulse, allowing it to align with the generated music, especially when processed through our two-stage *BNMusic* method.

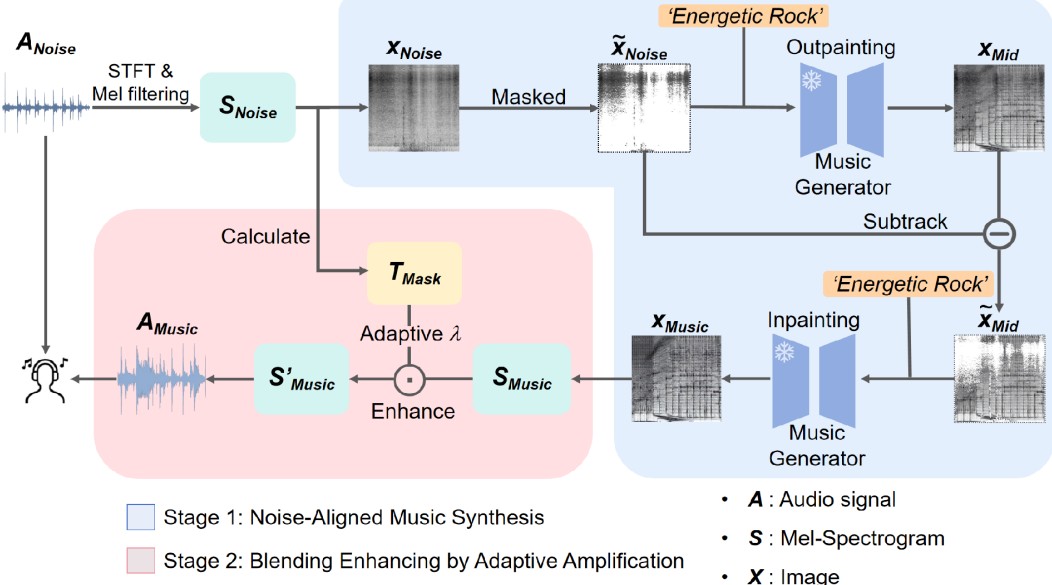

Figure 2: **Overall pipeline of our proposed *BNMusic* framework to achieve noise blending with frozen music generators.** The two stages of our approach are marked with different background colors. In Stage 1, our approach generates music that aligns with the noise, and in Stage 2 we adaptive amplify the music signal to reach the most ideal and reasonable blending with the noise.

**Pre-processing.** As the input noise $A_{\text{Noise}} \in \mathbb{R}^{t \times f_s}$, where $t$ is the length of audio in seconds and $f_s$ is the sampling rate (i.e., number of samples per second), is given to the system, we convert it from an audio signal into a mel-spectrogram $\mathbf{S}_{\text{Noise}} = \text{Mel}|\text{STFT}(A_{\text{Noise}})| \in \mathbb{R}^{W \times H}$, where Mel stands for the Mel-filtering process and STFT means Short Time Fourier Transform. Through this conversion, the input one-dimensional signal $A_{\text{Noise}}$ can be represented in a two-dimensional matrix, namely the mel-spectrogram $\mathbf{S}_{\text{Noise}}$. Once the mel-spectrogram is obtained, its amplitude values are mapped to grayscale pixel intensities in the range $[0, 255]$. In this mapping, lower pixel values represent louder regions, while higher values indicate quieter ones. This process yields a detailed grayscale mel-spectrogram plot, denoted as $\mathbf{x}_{\text{Noise}} \in [0, 255]^{W \times H \times 1}$, where the last dimension represents a single grayscale channel. With such a more perceptually relevant representation of the signal, the evolution of noise frequency content over time becomes easier to analyze and extract. After obtaining $\mathbf{x}_{\text{Noise}}$, we then apply a binary mask $\mathbf{M} \in \{0, 1\}^{W \times H}$ that masks out the area representing the higher energy part of the noise. Since frequencies with higher local energy are typically more perceptually salient and more likely to interfere with auditory perception, isolating these regions is crucial for subsequent blending. The masked image can be calculated as $\widetilde{\mathbf{x}}_{\text{Noise}} = \mathbf{x}_{\text{Noise}} \odot \mathbf{M} \in [0, 255]^{W \times H \times 1}$, where $\odot$ represents element-wise multiplication.

**Stage 1: Noise-aligned music synthesis.** We adopt a two-step outpainting and inpainting process to guide the generation of music around and within the high-energy regions of the noise spectrogram. This design enables the model to capture and preserve the noise's intrinsic rhythmic cues while diffusing them into a coherent musical structure, thereby ensuring that the generated music naturally aligns with the most perceptually salient parts of the noise. First, the mask $\mathbf{M}$ isolates the core noise region in the image $\widetilde{\mathbf{x}}_{\text{Noise}}$, dividing it into two parts. During the outpainting stage, the core noise area is preserved, and music is generated to fill the surrounding space, allowing the core information to diffuse outward. The masked mel-spectrogram plot $\widetilde{\mathbf{x}}_{\text{Noise}}$ and the text prompt $C_{\text{text}}$ are encoded into latent representations, then fed into the LDM [32] from Riffusion [6] for music generation, which is a modified version of Stable Diffusion-v1-5 fine-tuned for generating music's mel-spectrogram plot. Given a noisy latent representation $\mathbf{z}_t$ at timestep $t$, the model predicts the added noise $\boldsymbol{\epsilon}_\theta$ using a U-Net conditioned on both the corrupted mel-spectrogram $\widetilde{\mathbf{x}}_{\text{Noise}}$ and the associated text prompt $C_{\text{text}}$. Using the predicted noise $\boldsymbol{\epsilon}_\theta$, the posterior distribution of the previous latent state $\mathbf{z}_{t-1}$ is computed as:

$$p(\mathbf{z}_{t-1} \mid \mathbf{z}_t, \widetilde{\mathbf{x}}_{\text{Noise}}, C_{\text{text}}) = \mathcal{N}\left(\mathbf{z}_{t-1}; \mu(\mathbf{z}_t, \boldsymbol{\epsilon}_\theta(\mathbf{z}_t, t, \widetilde{\mathbf{x}}_{\text{Noise}}, C_{\text{text}})), \sigma_t^2 \mathbf{I}\right). \tag{1}$$

Here, $\mu$ is an analytically derived function that depends on $\mathbf{z}_t$. $\epsilon_\theta$ represents the predicted noise, $t$ denotes the diffusion step, and $\sigma_t^2$ is determined by a fixed noise schedule. The reverse process proceeds iteratively until $t = 0$, yielding the final latent $\hat{\mathbf{z}}_0$. This latent representation is then passed through the decoder $D$ to reconstruct the mel-spectrogram, including the previously masked region $\mathbf{x}_{\text{Mid}} = D(\hat{\mathbf{z}}_0)$.

The outpainted region of the $\mathbf{x}_{\text{Mid}}$ aligns the rhythmic patterns with the remaining noise content, leading us to invert the mask $\mathbf{M}$ to let the model inpaint on the area that represents the higher energy components of the noise:

$$\widetilde{\mathbf{x}}_{\text{Mid}} = \mathbf{x}_{\text{Mid}} \odot (\mathbf{1} - \mathbf{M}) \in [0, 255]^{W \times H \times 1}. \tag{2}$$

After a second round of inpainting on the inversely masked area on $\widetilde{\mathbf{x}}_{\text{Mid}}$, we obtain an image $\mathbf{x}_{\text{Music}}$, an image representing the complete musical content. This iterative process extracts rhythmic patterns from the noise and integrates them into the music, replacing noisy elements while eliminating the most distracting parts. The second round of inpainting is crucial for refining the composition, ensuring that the diffused information is reintegrated into the core, ultimately reconstructing a piece of music that not only preserves the rhythmic characteristics of the original noise but also aligns with its dominant perceptual features, thereby laying a solid foundation for effective blending in the subsequent stage.

**Stage 2: Blending enhancing by adaptive amplification.** To further enhance blending, we adaptively applies minimal loudness amplification, exploiting the music's alignment with the noise's high-energy regions to achieve effective auditory masking with modest gain, thereby reducing perceptual intrusiveness without disrupting the overall acoustic balance. To calculate the masking threshold, we first compute the noise's magnitude spectrogram using the Short-Time Fourier Transform (STFT), which extracts the magnitude of the complex STFT coefficients. The resulting matrix, $\hat{\mathbf{S}}_{\text{Noise}}$, is real-valued and belongs to $\mathbb{R}^{\hat{W} \times H}$, where $\hat{W}$ is the number of frequency bins and $H$ is the number of time frames. Based on previous research [28], the minimum signal-to-mask ratio (SMR) for Tune-Masking-Noise cases is typically between 21–28 dB. Using the minimum value of 21 dB, we derive the threshold matrix as:

$$\mathbf{T}_{\text{Mask}} = \text{Mel}|10^{\frac{20 \cdot \log_{10}(\hat{\mathbf{S}}_{\text{Noise}}) + 21}{20}}| \in \mathbb{R}^{W \times H}. \tag{3}$$

These thresholds $\mathbf{T}_{\text{Mask}}$ indicate that any sound exceeding these values will trigger auditory masking for the corresponding frequency and time. This allows us to determine when and how sounds can mask the noise. We then use gradient descent to find an optimal amplification factor $\lambda$, ensuring maximal auditory masking while keeping the total loudness of the music within a reasonable range. The amplified music signal can be represented as $\mathbf{S}'_{\text{Music}} = \mathbf{S}_{\text{Music}} \cdot \lambda \in \mathbb{R}^{W \times H}$, where $\cdot$ denotes scalar multiplication. Therefore we put up with an optimization function:

$$\lambda^* = \arg\min_\lambda \left\{ \text{SUM}(\alpha \cdot \mathbf{S}'_{\text{Music}}) + \text{SUM}(\max[(\mathbf{T}_{\text{Mask}} - \mathbf{S}'_{\text{Music}}) \odot \mathbf{M}, \mathbf{0}]) \right\}. \tag{4}$$

The formula optimizes the parameter $\lambda$ to minimize the objective function, finding the optimal solution $\lambda^*$. $\text{SUM}(\alpha \cdot \mathbf{S}'_{\text{Music}})$ represents the sum weighted music signal, where $\alpha$ controls the weight of the music signal in the optimization. $\text{SUM}(\max[(\mathbf{T}_{\text{Mask}} - \mathbf{S}'_{\text{Music}}) \odot \mathbf{M}, \mathbf{0}])$ is the sum of the noise masking term. This formula ensures amplification occurs only when the masking effect on the core area outweighs the global mel-spectrogram increase, maintaining a balance between enhancing core masking and avoiding unnecessary global amplification.

The amplified mel-spectrogram $\mathbf{S}'_{\text{Music}}$ represents the final music output we seek to use for masking the noise. We transform it back into an audio signal $A_{\text{Music}}$ using the following process:

$$A_{\text{Music}} = \text{ISTFT}(\text{Griffin-Lim}(\text{Mel}^{-1}(\mathbf{S}'_{\text{Music}}))) \in \mathbb{R}^{t \cdot f_s}, \tag{5}$$

where Griffin-Lim [30] is used to estimate and recover the phase information of the signal before performing the ISTFT (Inverse Short-Time Fourier Transform). $\text{Mel}^{-1}$ represents the inverse Mel-filtering operation. This approach reconstructs the audio signal from the generated mel-spectrogram.

## 4 Experiment

### 4.1 Experiment setup

**Dataset.** Our dataset consists of three components: noise clips, real music clips, and text prompts. To ensure diverse noise conditions, we source 1,000 segments from the **EPIC-SOUNDS** dataset [10],

covering 58 human action categories and 140 object types, and 300 additional segments from the **ESC-50** dataset [31], which includes 50 categories of real-world sounds such as thunderstorms, sea waves, and chirping birds. Together, these samples span a wide range of everyday acoustic environments and a broad frequency spectrum from 100 Hz to 10,000 Hz. For music data, we use 5,000 five-second clips from the **MusicBench** dataset [19], derived from 3,413 high-quality tracks across various genres, styles, and instrumentation. These serve as both ground truth and baselines for evaluation. Additionally, we construct a prompt set of 100 text descriptions across seven music genres via LLMs [27], covering *Pop*, *EDM*, *Rock*, *Hip-hop*, *Punk*, *Jazz*, and *Classical*. By pairing each noise clip with multiple prompts, we generate 14,200 music clips using two controllable generation models: **Riffusion**[6] and **MusicGen**[3]. Specifically, we pair each of the 1,000 EPIC-SOUNDS noise clips with five different prompts, and each of the 300 ESC-50 clips with seven prompts, resulting in 5,000 and 2,100 generated music pieces, respectively.

**Baselines.** We compare our results with three baseline methods. The first two use Riffusion's audio-to-audio generation [6] and MusicGen's melody-conditioned generation [3], both generated with the same noise and text prompt pair used in our method. The third baseline involves randomly selected real music from MusicBench [19]. All generated music clips are overlaid with their corresponding noise segments to simulate the actual auditory experience as perceived by users. This enables both objective and subjective evaluation of how well the music blends with environmental noise in realistic listening conditions.

**Implementation details.** Since loudness plays a critical role in masking perception, our approach aims to blend noise into music without relying on excessive volume, necessitating loudness normalization to ensure a pleasant and balanced auditory experience. To address this, we apply Pyln-norm [37], which uses the ITU-R BS.1770-4 model for loudness normalization, ensuring all audio clips are adjusted appropriately. The noise is consistently normalized to -18 dB LUFS in all evaluations. The Riffusion model [6] was used with default settings to ensure compatibility, and each sample was processed in approximately 5 seconds on an Nvidia 4090 GPU. The entire process takes approximately 0.28 seconds for preprocessing and amplification, with the majority of time spent on the two-stage generation, while system-induced delay remains minimal. The overall music signal control parameter, $\alpha$, was set to 0.14 to ensure adaptive amplification remained within a reasonable range. For evaluation, we compare our approach by overlaying music and noise clips to simulate real-world scenarios. In the real music baseline, half of the clips were real music paired with noise, and the other half served as ground truth for evaluation.

## 4.2 Evaluation

**Objective evaluation.** We use Fréchet Audio Distance (FAD) [11] and Kullback-Leibler (KL) divergence as our primary objective metrics. FAD measures how closely generated audio matches reference audio in terms of statistical properties, while KL divergence compares the probability distributions of generated and reference clips. Lower values of both metrics indicate greater similarity. FAD is computed over feature distributions across batches, while KL divergence is calculated pairwise between generated and reference audio clips. In our experiment, FAD and KL scores were calculated by comparing the combined noise and music audio to the real music ground truth. We also evaluate these metrics on both direct outputs and those normalized to match the noise's loudness. The objective evaluation results in Tab. 1 indicate that our method achieves the best FAD and KL scores across both scenarios, demonstrating effective blending with environmental noise, even when music and noise are presented at equal loudness. The low scores indicate that the combined audio is statistically and perceptually similar to real music, suggesting that distracting noise components are more effectively masked. This supports the effectiveness of our approach in enhancing the auditory experience by rhythmically and structurally aligning music with noise.

**Subjective evaluation.** Blending noise with generated music to enhance harmony is inherently subjective, as perceptions of auditory harmony can vary between individuals. To evaluate our approach, we conduct human evaluations with 50 samples, each containing five audio clips: the original noise, our result, and three adaptive amplified baselines. To closely approximate real-world conditions, all music clips are mixed with the corresponding noise prior to playback. Adaptive amplification is applied to both our music and the baselines for fair comparison. Testers first listen to the original noise alone, and then evaluate the mixed samples (i.e., our result and three adaptive amplified baselines), then score each on **OVL** (overall quality) and **PER** (perceived noise level) using

Table 1: **Objective evaluation results.** This table reports the Fréchet Audio Distance (FAD) and Kullback-Leibler (KL) Divergence scores tested on two scenarios—Loudness Normalized (left) and Direct Outputs (right)—on the EPIC-SOUNDS [10] and ESC-50 [31] datasets. As shown, our BNMusic method consistently achieves the best performance, highlighting the robustness and generalizability of our approach across different acoustic settings and datasets.

| Methods | Loudness Normalized | | | | Direct Outputs | | | |
| | EPIC-SOUNDS [10] | | ESC-50 [31] | | EPIC-SOUNDS [10] | | ESC-50 [31] | |
| | FAD↓ | KL↓ | FAD↓ | KL↓ | FAD↓ | KL↓ | FAD↓ | KL↓ |
|---|---|---|---|---|---|---|---|---|
| Noise Only | 34.17 | – | 27.39 | – | 34.17 | – | 27.39 | – |
| Random Music | 14.22 | 2.22 | 8.45 | 2.49 | 15.41 | 2.38 | 8.32 | 2.61 |
| MusicGen [3] | 13.28 | 2.14 | 8.62 | 2.43 | 10.95 | 1.85 | 7.74 | 2.33 |
| Riff A2A [6] | 20.06 | 2.90 | 12.62 | 3.26 | 13.15 | 2.25 | 9.11 | 2.70 |
| **BNMusic (Ours)** | **12.86** | **2.03** | **8.09** | **2.38** | **7.98** | **1.67** | **6.76** | **2.14** |

Table 2: **Subjective and objective evaluation on adaptively amplified samples.** We report subjective scores for overall quality (OVL) and perceived noise level (PER), along with objective metrics: Fréchet Audio Distance (FAD) and Kullback-Leibler Divergence (KL) evaluated on adaptive amplified audio samples. BNMusic achieves the highest OVL and PER scores and the lowest KL among all comparing methods, demonstrating its effectiveness in blending with noise and reducing the perceptual impact of background interference.

| Methods | Subjective Metrics | | Objective Metrics | |
| | OVL↑ | PER↑ | FAD↓ | KL↓ |
|---|---|---|---|---|
| Random Music | $2.93 \pm 0.58$ | $2.63 \pm 0.53$ | **6.84** | 2.07 |
| MusicGen [3] | $2.97 \pm 0.34$ | $2.68 \pm 0.54$ | 7.08 | 1.75 |
| Riff A2A [6] | $2.95 \pm 0.60$ | $3.24 \pm 0.67$ | 12.82 | 2.33 |
| **BNMusic (Ours)** | $\mathbf{3.67 \pm 0.55}$ | $\mathbf{3.84 \pm 0.63}$ | 7.98 | **1.67** |

a 1 to 5 Likert scale, where 5 indicates the most pleasant experience or the least perceived noise. This evaluation provides valuable insights into user perceptions of our method's effectiveness. More details and a sample of the user questionnaire is included in the Appendix C. As shown in Tab. 2, subjective evaluation results indicate that most users find our *BNMusic* segments provide the best hearing experience alongside environmental noise, outperforming all other baselines. Riffusion's Audio-to-Audio approach [6] ranks second in noise suppression but compromised musicality, as its outputs closely mimic the noise structure. MusicGen [3] and Real Music achieve lower PER scores, with MusicGen offering marginal noise masking through melody-aware generation, but still falling behind BNMusic and Riffusion.

**Visualization-based comparison.** To further illustrate the blending behavior, Fig. 3 presents several representative examples. Each group consists of five plots: the mel-spectrogram of a noise sample, followed by four heatmaps showing the difference between that noise and four types of music: Random Music, MusicGen, Riffusion-A2A, and our *BNMusic*. All music samples are loudness-normalized to match the noise before computing the difference to ensure a fair comparison. The heatmaps visualize the energy difference between the music and the noise. Red indicates positive differences, blue indicates negative, and darker colors represent larger magnitudes. These maps reveal how closely each music sample aligns with the noise in terms of spectral energy distribution. Among the four, Random Music shows the largest mismatch with the noise, especially in less active frequency bands. MusicGen also differs notably, but to a lesser extent. In contrast, Riffusion-A2A and our *BNMusic* demonstrate much closer alignment to the noise across the frequency spectrum. Their differences are more evenly distributed and less extreme, indicating better spectral blending. This suggests that A2A and *BNMusic* are more effective in matching the energy profile of the noise, which may underlie their superior auditory integration. However, both FAD and subjective evaluation results confirm that our *BNMusic* significantly outperforms A2A in terms of pleasantness and harmony.

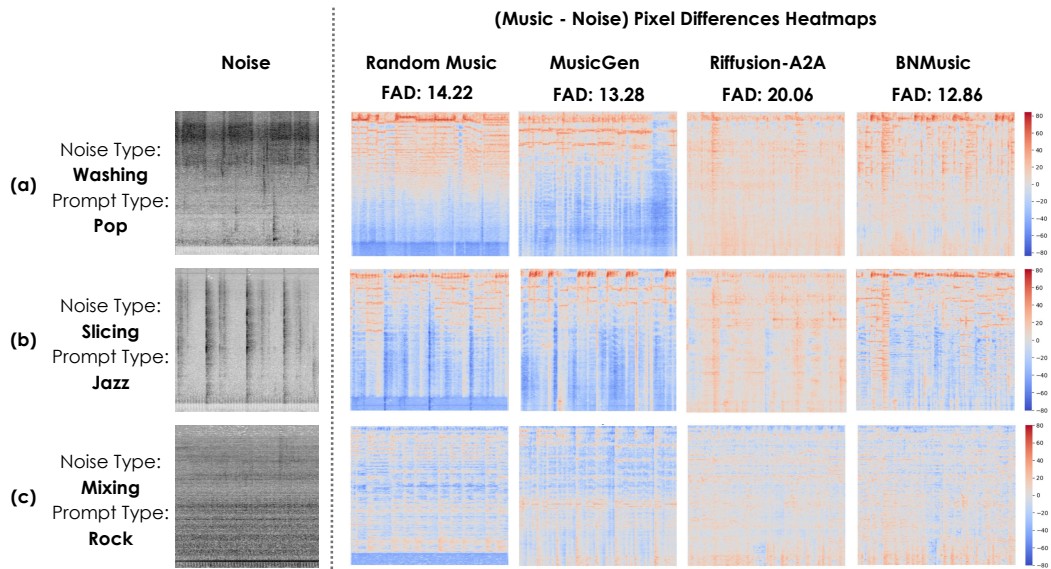

Figure 3: **Visualization of noise-music blending effectiveness across methods.** The left images display the mel-spectrograms of three types of noise, while the right heatmaps show the differences between the generated music and the noise. The heatmaps illustrate how four music samples blend with the respective noise. Red represents positive values, blue indicates negative values and darker colors correspond to larger magnitudes, highlighting the blending effectiveness of each music type.

Table 3: **Ablation study of method components.** We evaluate different combinations of method components, and report Fréchet Audio Distance (FAD) and KL Divergence (KL).

| Method Components | | | Metrics | |
|---|---|---|---|---|
| Outpainting | Inpainting | Adaptive Amplification | FAD↓ | KL↓ |
| × | × | × | 34.17 | – |
| ✓ | × | × | 8.68 | 1.89 |
| ✓ | × | ✓ | 9.18 | 1.84 |
| ✓ | ✓ | × | 8.00 | 1.78 |
| ✓ | ✓ | ✓ | **7.98** | **1.67** |

**Ablation study.** To further evaluate the contribution of each component in our framework, we conduct an ablation study by systematically disabling parts of our pipeline. Our method consists of three key components: Outpainting, Inpainting in Stage 1, and Adaptive Amplification in Stage 2. We test variants where one or more of these components were removed. As shown in Tab. 3, each component contributes to the final performance, and the full system, which combines outpainting, inpainting, and adaptive amplification achieves the best objective results. This highlights the importance of all three stages in enabling effective noise-aware music blending. As the results shown in Tab. 2, MusicGen [3] performs better in FAD due to amplification making the music more prominent. However, Random Music performs poorly in KL divergence as amplification fails to balance noise-music interaction. Our method, with pairwise adaptive amplification, aligns music and noise effectively and achieves better blending and a more pleasant auditory experience. This is consistent with the user study results, Random Music and MusicGen [3], despite higher music quality, still fail in blending effectively with the noise and making it less annoying. In contrast, our method ensures seamless blending and provides better auditory experience.

## 5 Conclusion and discussion

In conclusion, our *BNMusic* demonstrates superior performance in blending music with environmental noise compared to other methods, effectively reducing the annoyance of the noise while enhancing

the overall auditory experience. Through a series of experiments and ablation studies, we show the effectiveness of our approach, as well as the contribution of each key modeling component. By finding an optimal balance between maximizing the pleasantness of the music, controlling its loudness, and aligning it with the noise for more seamless blending, our method ensures that the combined sound provides the best listening environment.

**Limitations.** When users deliberately provide prompts that are poorly matched to the noise the system struggles to generate a coherent blend, and the influence of the prompt becomes less pronounced. This highlights the need for prompt-noise compatibility to achieve effective blending. While mel-spectrogram representations help reduce inference costs by compressing audio, the conversion between time and frequency domains introduces distortion that slightly affects music quality. Real-time processing is currently limited by slow generation speed, making it infeasible at this stage. Our primary focus is to demonstrate the effectiveness of the proposed method. For repetitive noise scenarios, practical applications can still be developed through offline recording and post-processing. Looking ahead, integrating our approach with faster generation techniques may enable near real-time performance and broaden the range of potential applications.

**Broader impacts.** This work aims to enhance auditory comfort in noisy environments by blending music with ambient noise in a perceptually harmonious way. It has potential applications in public transport, offices, etc, offering a more pleasant listening experience. However, if overused or applied without user control, such blending systems might unintentionally mask important environmental sounds or lead to listener fatigue over time.

## Acknowledgments and Disclosure of Funding

This paper is co-supervised by Prof. Ye Zhu and Prof. Yu Wu. This work was partially supported by the National Natural Science Foundation of China under grant 62372341 to CZ, HH and YW. This research was also partially supported by an Amazon Research Award to YZ. Any opinions, findings, and conclusions or recommendations expressed in this material are those of the author(s) and do not necessarily reflect the views of Amazon. YZ also acknowledges the travel funding support by the French National Research Agency (ANR) via the "GraspGNNs" JCJC grant (ANR-24-CE23-3888), coordinated by Johannes F. Lutzeyer from École Polytechnique.

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

## Technical Appendices and Supplementary Material

This appendix provides additional details to complement the main text and support a deeper understanding of our work. Sec. A investigates how the FAD score varies with changes in loudness. Sec. B elaborates on the details of Stage 1 in the method described in Sec. 3, including a visual representation of the generation process. Sec. C supplements the subjective evaluation by offering further explanation and presenting a representative sample.

## A    Loudness-FAD relationship

We observed that loudness has a noticeable impact on the Fréchet Audio Distance (FAD)[11], which evaluates how closely generated audio resembles reference ground truth music in terms of statistical properties. To investigate this effect, we conducted a dedicated experiment examining how FAD varies with changes in loudness. Specifically, we normalized the loudness of outputs from three baseline methods and our approach to a range between -24 and -3 dB LUFS, and then calculated the FAD scores for each loudness level. As shown in Fig.4, the results consistently reveal that the FAD score reaches its minimum—indicating the best match to real music—when the loudness is between -18 and -12 dB LUFS. Interestingly, this range coincides with the dominant loudness levels of the reference ground truth, which likely reflects the most comfortable listening range for the human ear. This experiment also suggests that while higher loudness can enhance the masking effect of noise, excessively high levels can degrade perceptual quality. Therefore, we aim to keep the overall loudness within an appropriate range to achieve better blending and provide the most comfortable experience for the listener.

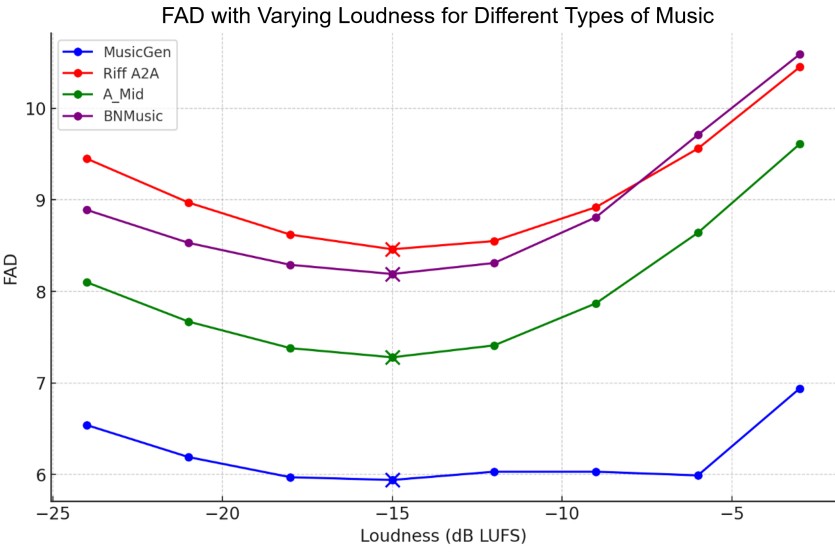

Figure 4: The lowest FAD values for each type of music are highlighted, and they all appear in the -15 dB LUFS, indicating that for all types of music in our experiment, an optimal loudness level consistently falls between -18 to -12 dB LUFS.

## B    More details in Stage 1

This section provides a detailed supplement to Stage 1 of the method described in Sec. 3. Through visualizations, we illustrate the motivation for enforcing rhythmical alignment and explain how our Stage 1 achieves this alignment via a combination of outpainting and inpainting. This approach not only ensures temporal coherence but also preserves sufficient musicality in the generated content. Additionally, we describe the strategy used to select the core area of the noise mel-spectrogram plot, which serves as the foundation for the alignment process.

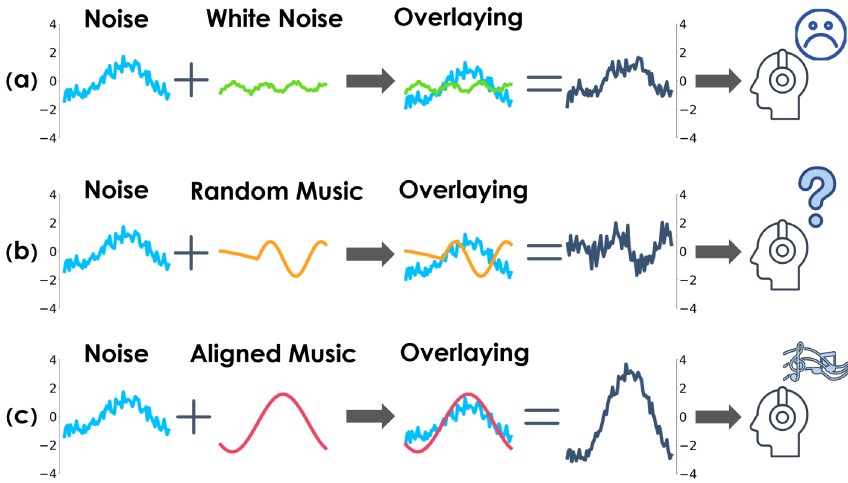

Figure 5: Illustration of how conventional auditory masking, often using overly low volume and mismatched rhythm, can disrupt the listening experience. Proper volume adjustment and rhythmic alignment are essential for achieving a more harmonious and pleasant blend with background noise.

**The significance of alignment.** Our approach leverages generative models to produce music that rhythmically aligns with the background noise. This alignment facilitates natural blending, as illustrated in Fig. 5, reducing potential conflicts between the two sources. By ensuring temporal coherence from the outset, the combined audio avoids introducing additional disturbances into the soundscape. As a result, the subsequent adaptive amplification can be applied more conservatively while still achieving effective masking. Even in frequency regions where complete masking is not possible, the improved alignment enhances perceptual harmony and helps minimize the listener's awareness of the underlying noise.

**The significance of all steps.** Fig. 6 illustrates the detailed process of transforming a single noise sample, represented as $\mathbf{x}_{\text{Noise}}$, into the final output music in an image representation, $\mathbf{x}_{\text{Music}}$. As shown in Fig. 6, the outpainting step primarily focuses on diffusing information from the preserved core region of the noise $\widetilde{\mathbf{x}}_{\text{Noise}}$ into the surrounding areas. This diffusion embeds contextual information into the surrounding music during the generation of $\mathbf{x}_{\text{Mid}}$. However, at this stage, directly converting $\mathbf{x}_{\text{Mid}}$ into an audio signal $A_{\text{Mid}}$ would retain noise content from the core region, significantly degrading the listening experience. To address this, a subsequent inpainting step is required to mask the remaining core noise area and replace it with structured, harmonious music that aligns with the text prompt. During this inpainting process, the information previously embedded in the surrounding music during outpainting diffuses back into the core region, ensuring seamless integration. The result, $\mathbf{x}_{\text{Music}}$, represents a cohesive and complete musical piece.

Furthermore, as demonstrated in Fig. 7, both our approach and the Riffusion's audio-to-audio generation [6] exhibit the most effective alignment with the noise. The results of our approach, the Riffusion [6]'s, and the MusicGen [3]'s are all generated conditioned on the noise, expected to maintain a strong rhythmic consistency for a more seamlessly blending. In contrast, the result of MusicGen's melody-conditioned generation [3], as well as the randomly chosen music, fail to achieve similar rhythmic synchronization with the noise, as expected. This highlights the superior ability of our method to align the generated music with noise, making it more coherent and seamlessly integrated while maintaining pleasant to the ear.

**The strategy of picking thresholds.** The selection of 10%–20% of the area with smaller pixel values as the mask region is based on empirical observations. Since the mask is extracted pixel-wise with a value range of 0–255 while the smaller pixel value indicates the higher energy level of the mel-frequency, small variations in pixel intensity can lead to significant differences in the mask area, especially in images with relatively low contrast. Our goal is to ensure that the mask region captures the primary high-energy frequency areas while keeping its size minimal. This approach provides the model with greater flexibility to generate the desired musical elements. Conversely, during the

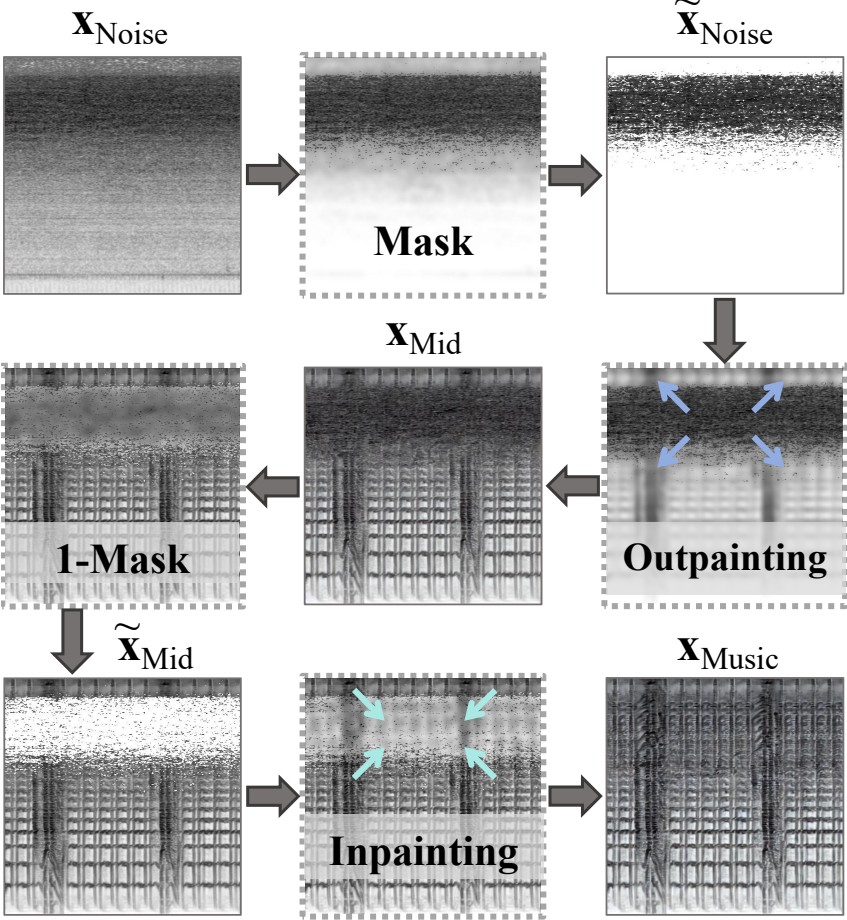

Figure 6: The more detailed illustration for the generation process in Stage 1, transitioning from $\mathbf{x}_{\text{Noise}}$ to $\mathbf{x}_{\text{Music}}$, should emphasize the distinct processing stages and the regions primarily affected during each step. This would include highlighting how the inpainting step serves as the pivotal transformation within the process, where chaotic noise regions are replaced with structured and meaningful music content.

inpainting phase, the preserved core region may sometimes occupy a relatively small proportion of the overall area. In such cases, the limited space can make it challenging for inpainting to generate sufficiently detailed and coherent musical content. To address this, we adjust the threshold to slightly enlarge the mask area, enabling the generation of a more complete, harmonious, and cohesive musical result.

## C  More details about subjective evaluation

This section provides additional details regarding the subjective evaluation process. It outlines the evaluation protocol and criteria used to assess perceptual quality, and includes a sample of the questionnaire presented to listeners during the study.

The participants would be seeing these words:

OVL (Overall): Measures the overall quality and pleasantness of the audio.

Perceptibility: Indicates how noticeable the original noise is in the presence of the music.

Both metrics are rated on a scale from 1 to 5, where 1 represents the least pleasant sound or the noise being most perceptible, while 5 denotes the most pleasant sound or the noise being least perceptible.

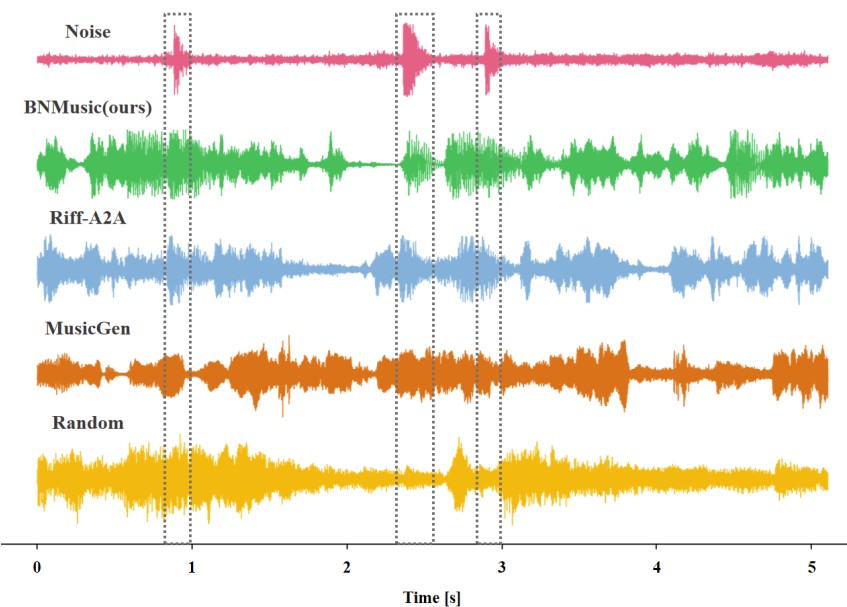

Figure 7: The waveforms of a set of samples, consisting of noise, a random music track, and three music segments generated based on the noise, are shown. As highlighted, our method achieves one of the best alignment effects, where any impulsive sound from the noise is seamlessly blended with a corresponding strong musical sound, ensuring a smooth integration between the two.

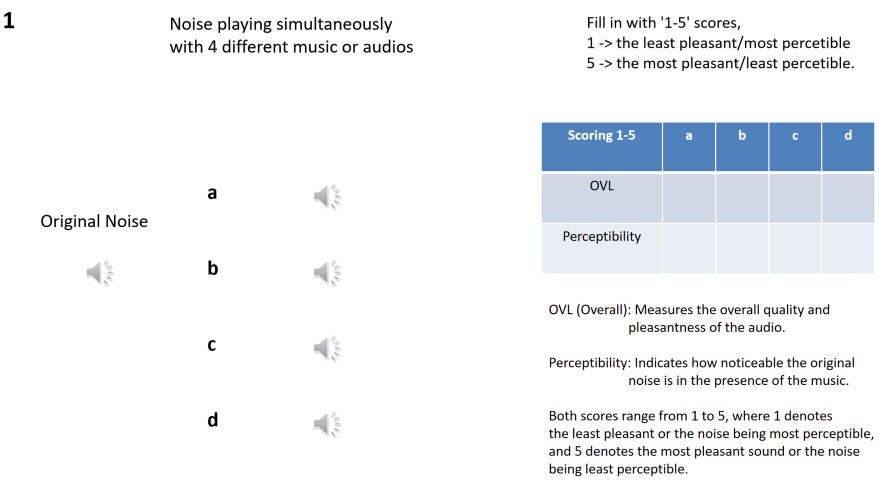

Figure 8: A sample page of the subjective evaluation.

Each participant was presented with a set of audio clips, including the original noise, three music clips generated using the noise, and a randomly selected real music piece all overlaid with the noise. The participants were asked to rate the overall quality and perceptibility of each clip. A sample page of the questionaire is presented in Fig. 8

