# OpenReview forum: "BNMusic: Blending Environmental Noises into Personalized Music"
_NeurIPS.cc/2025/Conference — NeurIPS 2025 poster_

### Official Review · Reviewer_TZwc · 2025-06-06

**Clarity:** 4
**Significance:** 3
**Originality:** 3
**Rating:** 5
**Confidence:** 2

**Summary:**

This work addresses the task of acoustic masking to reduce the impact of environmental noise. Unlike conventional approaches that rely on active noise cancellation, the authors propose a novel and interesting solution: generating music from the environmental noise itself. The proposed framework uses a diffusion-based music generator to synthesize music conditioned on the input noise and text prompts, which is then blended with the original noise signal to achieve acoustic/noise masking. The evaluation includes quantitative analysis using defined metrics, human evaluation, and qualitative assessment through visualizations of the differences between the generated music and the original noise. An ablation study is conducted to show the contributions of key components in the framework. The supplementary material includes audio demos showcasing music generated from environmental noise.

**Questions:**

My questions are included in the Strengths and Weaknesses section. I expect the authors' rebuttal to address these points.

**Ethical Concerns:**

["NO or VERY MINOR ethics concerns only"]

**Final Justification:**

The authors’ rebuttal addressed my concerns, and I will maintain my current rating.

**Limitations:**

Yes

**Paper Formatting Concerns:**

No concerns

**Quality:**

3

**Strengths And Weaknesses:**

Strength:
- The paper is well-written and easy to follow. Most of the questions I had while reading were clarified by the end.
- The proposed task is novel and interesting, and the authors provide a clear and compelling motivation for it.
- The experimental results demonstrate that the proposed method performs strongly compared to relevant baselines. The choice of baselines is reasonable, and it is particularly interesting to see that using generated music significantly outperforms using random music.

I did not identify any major weaknesses in the paper. However, there are a few points that may benefit from further clarification:
- In Stage 1, how is the core noise area selected? Is the final output sensitive to this selection?
- Regarding the objective evaluation: what is considered the ground-truth music? Since the output music is generated based on both the noise and a text prompt, it’s unclear how ground-truth references are defined or obtained. I may be missing something here.
- In the qualitative evaluation (Figure 3), Riffusion-A2A appears to produce better results than BNMusic. Why then does Riffusion-A2A perform significantly worse in the quantitative results shown in Tables 1 and 2?
- In the supplementary video, the generated music clips are quite short. Is it feasible to use longer noise segments to generate extended music tracks? In real-world scenarios, environmental noise often changes over time, rather than repeating the same pattern.

---

> ### Author Rebuttal · Authors · 2025-07-30
>
> ## Rebuttal for reviewer TZwc
>
> We thank the reviewer for the encouraging and constructive feedback. We appreciate the recognition of our work as *well-written*, the *novel* and *interesting* nature of the proposed task, the *clear* and *compelling motivation*, and the *strong performance* of our method against *reasonable baselines*, especially the finding that *generated music significantly outperforms random music*. Below, we address the reviewer’s questions and requests for clarification in detail.
>
> >Q1: In Stage 1, how is the core noise area selected? Is the final output sensitive to this selection?
>
> **A1:** Thank you for the thoughtful question.
> 1. In Stage 1, we select the top 10%–25% highest-energy regions of the mel-spectrogram as the conditioning mask. This design is grounded in psychoacoustic background: **energy-dominant components** are the most easily perceived by human listeners [1,2] and thus the most disruptive in noisy environments. By leveraging the auditory masking effect, our goal is to generate music that can perceptually mask these prominent noise components. Therefore, we use these high-energy regions both as conditioning input in Stage 1 to guide the music generation, and in Stage 2 to compute adaptive amplification weights for effective masking.
> 2. To assess the sensitivity of the output to the mask design, we conducted additional experiments with alternative masking strategies:
>
> - Randomly selected 20% region
> - Larger high-energy area (30%–50%)
> - Our original 10%–25% high-energy mask
>
>     The results are summarized below:
>
>     | Masking Strategy   | FAD ↓  | KL ↓   |
>     |--------------------|--------|--------|
>     | Random 20% region | 17.82   | 2.47   |
>     | Top 30%–50% high-energy area     | 13.79   | 2.14   |
>     | Top 10%–25% high-energy (ours) | 7.98   | 1.67   |
>
>     As shown, selecting a focused and moderate-sized high-energy region leads to better performance. Enlarging the area dilutes the effectiveness of the condition, and random selection fails to capture perceptually salient noise components. These results validate our design choice as both principled and empirically effective.
>
> >Q2: Regarding the objective evaluation: what is considered the ground-truth music? Since the output music is generated based on both the noise and a text prompt, it’s unclear how ground-truth references are defined or obtained. I may be missing something here.
>
> **A2:** Thank you for raising this important point. In our evaluation, the term “ground-truth” does not refer to a one-to-one matched target for each noise instance. Rather, it represents an idealized reference: in the best-case scenario, the blended result (noise + output music) should be perceptually indistinguishable from real, harmonious music, such that the presence of noise is no longer noticeable to the listener.
>
> Therefore, when computing objective metrics such as FAD or KL divergence, we use a collection of real music tracks from the MusicBench dataset as a reference set. The goal is not to evaluate whether the blended audio match any specific reference track, but rather to assess how closely it resembles natural music at a distributional and perceptual level. This allows us to measure the overall musicality and coherence of the output, without requiring one-to-one correspondence with the reference tracks.
>
> >Q3: In the qualitative evaluation (Figure 3), Riffusion-A2A appears to produce better results than BNMusic. Why then does Riffusion-A2A perform significantly worse in the quantitative results shown in Tables 1 and 2?
>
> **A3:** Thank you for pointing this out. The apparent discrepancy between the qualitative and quantitative results can be explained as follows:
>
> 1. Figure 3 in the original manuscript is designed to visualize the spectral difference between the generated music and the input noise, aiming to reflect **how closely the frequency distributions align**. In this regard, Riffusion-A2A indeed shows a smaller difference, which indicating that its output is more similar to the noise mel-spectrogram. However, this comes at a cost: the generated audio tends to **sound like a slightly more melodic version of the noise itself**, rather than coherent music.
>
> 2. This trade-off is also reflected in the quantitative results, where Riffusion-A2A shows lower similarity to real music and performs worse in terms of perceptual quality metrics. Its outputs are closer to noise than to music, which **contradicts** the main purpose of our work - to **blend noise with harmonically rich music and reduce listeners' perception of the noise**. As comparison, our method achieves a better balance between alignment with the noise and musicality, leading to higher perceptual quality and better blending.
>
>
> >Q4: In the supplementary video, the generated music clips are quite short. Is it feasible to use longer noise segments to generate extended music tracks? In real-world scenarios, environmental noise often changes over time, rather than repeating the same pattern.
>
> **A4:** Thank you for the insightful comment. While the base model Riffusion supports generating multiple 5-second segments to extend audio length, maintaining coherence across segments is challenging and often results in discontinuities in musical flow. In other words, **long-form music generation remains an open and active research challenge** in the field, particularly when aiming for thematic and structural consistency over time. In our case, this challenge is further compounded by the need to adapt to evolving noise. Improving long-range coherence is thus an important direction for future work, especially for real-world scenarios with dynamic environments.
>
> ---
>
> [1] Hugo Fastl and Eberhard Zwicker. Psychoacoustics: Facts and Models. Springer-Verlag, Berlin, Heidelberg. 2006.
>
> [2] Moore, B. C. J.. An Introduction to the Psychology of Hearing (6th ed.). Brill. 2012

---

> > ### Comment · Reviewer_TZwc · 2025-08-03
> >
> > I thank the authors for their rebuttal. It answered my questions, and I would like to keep my rating.

---

> > > ### Author Response · Authors · 2025-08-04
> > >
> > > We appreciate the reviewer’s valuable feedback, and we’re glad our responses were helpful in addressing your questions.

---

### Official Review · Reviewer_q6Wg · 2025-06-17

**Clarity:** 2
**Significance:** 2
**Originality:** 2
**Rating:** 5
**Confidence:** 4

**Summary:**

This paper introduces BNMusic, a novel framework for blending environmental noise into personalized, rhythmically-aligned music, as an alternative to traditional acoustic masking or noise cancellation.
Rather than suppressing noise, the method reduces its perceptibility by integrating it musically, offering a more comfortable auditory experience in shared spaces.
The approach relies on a two-stage spectrogram-level pipeline: (1) noise-aligned music generation via outpainting and inpainting using latent diffusion (Riffusion), and (2) adaptive loudness amplification optimized using psychoacoustic masking thresholds to ensure effective yet subtle blending.
Experiments on EPIC-SOUNDS and ESC-50 noise sources demonstrate quantitative and subjective improvements over Riffusion, MusicGen, and real music baselines in terms of perceptual quality and masking effectiveness.

**Questions:**

1. One big question is about the outpainting and inpainting process. From the Figure 6, when you only have a noise for example in the high frequency range, how can the model to outpaint the rest of the region that are related to the noise? These noise regions could be treated as conditional signal, but what is the relationship between these conditional signal with the outpainted regions? For example, how would you link the wind noise that could jam lots of frequency bands with any music signal? In the image inpainting or outpainting, the outpainted region could semantically align with the conditioned region. Can the author provide mroe justifications here?
2. Why do you choose to use Griffin-Lim algorithm rather than Vocoder for spectrogram to audio converting? It is quite common for audio generation based on the spectrogram to use vocoder to decode the spectrogram back to the audio rather than Griffin-Lim algorithm. What is the concern here?
3. Visualization issues in Fig3, if you want to directly visualize the sptrogram difference. Would calculate the absolute distance between the generated music and noise while visualizing with a better color scheme like "jet" be better?
4. Other music or audio generation work would also use IS, KL and other metrics to evaluate the performace, could you provide more analysis of why you didn't include this metrics or provide more comparison with different metrics? Using the Audioldm_eval github repo, you can get all of these metrics at once.

I would like to raise my score if my questions are well addressed.

**Ethical Concerns:**

["NO or VERY MINOR ethics concerns only"]

**Final Justification:**

I think most of my concern are addressed and I appreciated the author's discussio on the real-time application for the proposed method. And I think the overal idea of blending the music into noisy to surpass it is cute so I would increase my score to 5.

**Limitations:**

See the weaknesses.

**Paper Formatting Concerns:**

No such concern

**Quality:**

3

**Strengths And Weaknesses:**

Strengths:
1. The proposed framework and task is interesting. Blending noise into music and make it less noticeable is quite cute an idea.
2. Evaluation results show that the proposed method can achieve better noise blending performance in both subjective and objective metrics.

Weaknesses:
1. The biggest limitation and weakness is the real-time applicability. While I feel the generated audio is quite good, this is still a post-processing method that when you have already recorded the audio and then generate the corresponding music signal to blend the noise together. In the real world scenario, noise can be time-variant signal and it seems that these kind of system need to generate the blending music in real time to adapt to different noisy sound. Can you provide mroe analysis on this aspect?
2. The description on the second stage is a bit hard to understand. There are lots of equations and descrptions that are confusion. From my understanding, let A denote the nosie and B denote the generated music, and let $\lambda$ denotes the weights when you blend the music, i.e. A+$\lambda \cdot$B. If you want to totally get rid of the noise, theoretically you can make the $\lambda$ much larger than 1 which I understand is not possible in reality. However, if you put the regions in $\lambda$ that corresponding to the noise spectrogram close to 0 while mainting other regions (don't contain noise) to 1, the blended results would still be good right? Can you provide more explanations and justifications on this part?

---

> ### Author Rebuttal · Authors · 2025-07-30
>
> ## Rebuttal for reviewer q6Wg
>
> We thank the constructive and encouraging feedback, and appreciate the recognition of *our framework and task as interesting*, the characterization of our method as *quite a cute idea*, and that our evaluation *shows better noise blending performance in both subjective and objective metrics*. Below, we address the reviewer’s questions and concerns in detail.
>
> >Q1: Main limitation lies in real-time applicability—can the method adapt to time-varying noise instead of relying on offline post-processing?
>
> **A1:** Thank you for raising this important point. We elaborate on our analysis and discussions around different scenarios below.
>
> 1. Our method is well-suited for many real-world scenarios **where environmental noise exhibits repetitive or predictable patterns**, such as the operation of elevators, automatic garage doors, or air conditioning systems. In such cases, even though our current pipeline runs as a post-processing step, the **consistency and predictability of the noise** enable the system to **leverage prior statistical patterns**, allowing it to generate perceptually coherent and rhythmically aligned music. This makes the system effectively usable in **real-time or low-latency settings**, without requiring strict online adaptation at every moment.
> 2. As for **dynamic noise patterns**, our pipeline is theoretically capable of running with low latency. The current most time-consuming stage is music generation, which can be significantly accelerated using recent advancements in fast diffusion inference techniques, such as SoundCTM [1] and Presto [2]. According to the speed-up ratios reported in these papers, a 5-second audio clip can be generated in under 0.1 seconds on a single GPU. When combined with our preprocessing and Stage 2 adaptive amplification, both of which take approximately 0.3 seconds, the full pipeline can operate in under 0.5 seconds. With continues improvements in hardware and model optimization, this latency could be reduced even more, making the system viable for interactive or near-real-time applications where the noise evolves gradually. In future work, we plan to incorporate these acceleration strategies into our pipeline to support faster and more responsive performance.
>
> >Q2: Stage 2 blending seems unclear—why not simply reduce $\lambda$ to near zero in noisy regions for a good blending?
>
> **A2:** Thank you for the question. We would like to clarify our motivation behind the adaptive amplification strategy:
>
> 1. Our method is grounded in auditory masking theory. The goal is to use music with higher energy to mask the stronger components of noise, rather than expose them. By ensuring the musical signal **exceeds** the noise level in high-energy regions, we effectively suppress the perceptual salience of the noise. In contrast, setting λ close to zero would directly expose those noisy regions without masking, which goes against our core design principle, aiming to make noise less perceptible, not more exposed.
>
> 2. Blending does not mean fully exposing the noise. Our goal is to improve the auditory experience by **reducing the listener’s perception of the noise**, which is difficult to achieve if the noise is left entirely unmasked. Instead, we rely on auditory masking to suppress the most salient parts of the noise by amplifying well-matched musical content in those regions. This way, the listener perceives a more music-dominant and harmonious soundscape, where the noise is perceptually embedded rather than directly interfering. Reducing λ to near zero in noisy regions would result in the noise being fully exposed, which undermines the masking effect and ultimately degrades the overall perceptual quality.
>
>
> >Q3: How does the model outpaint meaningful music from noisy regions? What’s the semantic link between the masked input and the outpainted area? How to link noise that jam lots of frequency bands with any music signal?
>
> **A3:**
> 1. Our system is built upon a generative model trained to synthesize coherent and harmonious musical spectrograms. Much like how Stable Diffusion can generate a complete and semantically meaningful image from a partial visual region in inpainting or outpainting tasks [3,4,5]—such as generating a human body from a given face image—our model learns to infer the full musical structure based on a partial region and a text prompt. In our case, the high-energy noise region serves as a conditioning signal, and together with the prompt, guides the generation of the remaining music in a way that is both musically plausible and contextually coherent.
>
> 2. The relationship between the conditioned signals and the outpainted region is similar to image inpainting/outpainting tasks. The model is compelled to extract meaningful patterns (rhythm, texture, or dynamics) from the input region and use them to inform the surrounding generation. In other words, the conditioned region serves as a semantic and stylistic anchor that influences the generated music, ensuring the output aligns with the perceived "intent" of the original noise segment.
>
> 3. Our masking strategy focuses only on the top 10%–25% highest-energy regions in the spectrogram, rather than conditioning on all noisy areas. By targeting only the most perceptually dominant components, our method avoids being overwhelmed by the full extent of broadband noise. As a result, even when the noise jam lots of frequency bands, it does not adversely affect the generation process - only the most relevant acoustic features are used to guide the model, ensuring both control and robustness.
>
>
> >Q4: Why choose to use Griffin-Lim algorithm rather than Vocoder for spectrogram to audio converting?
>
> **A4:** We chose to use the Griffin-Lim algorithm for waveform reconstruction due to both practical and methodological considerations:
>
> 1. Our method is built upon the Riffusion model, which originally employs Griffin-Lim for audio reconstruction. To maintain consistency with the base model and avoid introducing additional confounding factors, we retained this default setting.
> 2. The mel-spectrogram configuration in Riffusion, such as mel scaling, hop size, is not directly compatible with standard neural vocoders like HiFi-GAN. Using such vocoders would require retraining or fine-tuning, which is beyond the scope of our current study.
> 3. Our primary focus is on improving the perceptual blending between noise and generated music, rather than optimizing audio fidelity.
>
> That said, we agree that incorporating a neural vocoder could have the potential to further improve the perceptual quality of the generated audio. This is a very helpful suggestion, and we will consider integrating a vocoder-based decoding pipeline in future revisions to enhance the overall listening experience.
>
> >Q5: Visualizing with color scheme like "jet" would be better.
>
> **A5:** Thank you for the suggestion. We agree that using the absolute difference with a better color scheme like jet can improve clarity. Due to the policy of rebuttal this year, we are not able to provide the figures but only text. We will update the visualization accordingly in the revision.
>
> >Q6: Why not include more metrics in Audioldm_eval?
>
> **A6:** Thank you for the suggestion.
> 1. In our paper, we primarily use FAD and KL divergence to evaluate our system. These metrics are computed on the **final blended outputs**, which **consist of the output music mixed with conditioned noise** and **represent what the listener would actually hear**. Ideally, our goal is to make the noise perceptually disappear and allow the listener to experience the audio as if it were real music. Both FAD and KL divergence are well-suited for this: they measure distributional distances between generated and real music from a group-wise (FAD) and pairwise (KL) perspective, thus capturing perceptual and statistical closeness to natural music.
> 2. Although *Audioldm_eval* includes a wide range of metrics, many of them are **tailored to general audio generation tasks** and are **not designed to evaluate perceptual masking effectiveness or contextual audio blending**. Specifically, metrics such as Inception Score (IS) and CLAP score are not directly appropriate for our use case. IS is intended to assess the diversity and clarity of generated audio, which is useful for general synthesis tasks but does not align with our objective of perceptual integration of music and noise. Similarly, CLAP measures text-audio alignment, which we do not explicitly optimize, as our model relies only on inherited text conditioning from the base model (Riffusion). Other common metrics face similar limitations and may yield evaluations that do not reflect the perceptual goals of our system.
> 3. Ultimately, this highlights a broader issue in the field: there is currently no widely accepted metric for quantifying the perceptual effectiveness of auditory masking or audio blending based on psychoacoustic principles. This is especially challenging when the goal is not to evaluate the audio itself in isolation, but its perceptual function in context. As such, we complement our evaluation with human listening studies, which offer a more direct and reliable reflection of perceptual quality.
>
> ---
>
> [1] SoundCTM: Unifying Score-based and Consistency Models for Full-band Text-to-Sound Generation. ICLR25.
>
> [2] Presto! Distilling Steps and Layers for Accelerating Music Generation. ICLR25.
>
> [3] Inpainting using Denoising Diffusion Probabilistic Models. CVPR2022.
>
> [4] SmartBrush: Text and Shape Guided Object Inpainting with Diffusion Model. CVPR2023.
>
> [5] Continuous-Multiple Image Outpainting in One-Step via Positional Query and A Diffusion-based Approach. ICLR 2024.

---

> ### Comment · Reviewer_q6Wg · 2025-08-05
>
> I would like to thank the authors for detailed response. Your replies have addressed most of my questions and concerns. My biggest concern about the real time processing is addressed. As you claimed, if a 5 second audio trunk could be process within 0.5 second of your whole pipeline, I would imagine some chunk based or stream based efficient model could be inplemented to boost the user's hearing experience in real time. I also encourage you to include more discussion on the evaluation metric for these kind of audio generation task since as you suggested there is not a common or standard metric to evaluate the perceptual effectives. Overall I like this cute idea of blending new audio contents to surpase the noisy signal. And I will raise my score to 5.

---

> > ### Author Response · Authors · 2025-08-06
> >
> > We thank the reviewer for the thoughtful feedback and encouraging words. We're pleased that our responses have helped clarify your concerns. We will integrate these points into the revision of the paper.

---

### Official Review · Reviewer_cD4J · 2025-06-29

**Clarity:** 4
**Significance:** 3
**Originality:** 3
**Rating:** 5
**Confidence:** 4

**Summary:**

This paper proposes a new insightful solution to the social issue of environmental noise in public areas by leveraging generative AI for music. Environmental noise in public areas is one of the major issues affecting the quality of human life.Traditionally, two main approaches have been employed. The first involves individuals using noise-reduction technology in headphones or earphones. However, this approach provides only limited solutions as it requires additional devices for each individual. The second approach involves generating different sounds by leveraging human psychological auditory characteristics to reduce the unpleasantness of the original noise. However, this method faces the challenge of requiring sounds with significantly high energy levels to effectively reduce noise impact. The authors propose a two-stage method consisting of (1) extracting the musical properties (primarily rhythm) of noise and generating music that matches these properties, and (2) equalizing the generated music based on human psychological auditory characteristics and mixing it with the noise. The authors comprehensively investigate the effectiveness of the proposed method from two perspectives: objective indicators and subjective evaluations.

**Questions:**

This paper proposes a very interesting new use case for generative AI. The paper is very well written and very easy to understand. In particular, the introduction speaks volumes about the appeal of this new application. I actually found myself asking as I was reading the paper, "How on earth can a music generation AI that takes into account psycho-aural characteristics be constructed?" and I was very much looking forward to the subsequent contents. Actually, generating music that prompts environmental noise and then suppresses it based on psycho-auditory characteristics seems like a truly magical tool. However, given its conceptually attractive goals, I get the impression that the actual system presented in this paper is somewhat more limited than I had hoped. My guess is that since it is very difficult to modify and improve the generation AI itself, the authors have chosen to use a fixed and pre-prepared generation AI and instead introduce outpainting and inpainting mechanisms to substitute a pseudo-ideal psycho-aural generation AI that takes into account ideal psycho-aural characteristics. I agree that this method of building the system is certainly one of the practical and reasonable solutions. This paper is totally unquestionable as a systems development paper. On the other hand, in a highly competitive and advanced technology presentation venue such as NeurIPS, this practical solution may have room for one more step of improvement in my subjective opinion. More specifically, it seems to me that building a generative AI that directly incorporates psychological auditory characteristics from its learning stage should have been the research subject that this paper was originally intended to address.I understand that these arguments of mine may perhaps seem unreasonable to the authors. So I would like to read the responses with an open mind, including the authors' rebuttals and the comments of other reviewers. Again, I thank the authors for sharing their very fascinating new perspective on the use of generative AI.

[Major concern 1: Validity of outpanting and inpainting mechanism instead of directly incorporating psyco-aural characteristics into generative AI]

- I am questioning whether it is possible to utilize the effects of acoustic masking based on psychological auditory characteristics at the stage of generating music.I understand that the effects of psychological auditory characteristics are utilized in Stage 2 of the authors' method. However, Stage 2 appears to retain only functions similar to an equalizer. To obtain generated music that utilizes psychological auditory characteristics as described in the introduction, it seems necessary to explicitly introduce these effects in Stage 1 of the authors' method.

[Major concern 2: Validity of prompt design (Mask M)]

- I do not fully understand the design guidelines for mask M, which is used to extract high-energy portions from noise. For example, if mask M is intended to extract rhythmic elements from noise, it seems possible to use a method that extracts periodic elements. Alternatively, if the goal is to extract frequency patterns that co-occur simultaneously from noise, it seems more appropriate to use a method that extracts low-rank structures from matrices, such as non-negative matrix factorization.I am unclear about the motivation behind the authors' decision to assign mask M the role of focusing solely on high-energy regions.

**Ethical Concerns:**

["NO or VERY MINOR ethics concerns only"]

**Final Justification:**

I am grateful for the authors' strong response and am both surprised and impressed. This is a very strong rebuttal. My concerns have been completely resolved beyond my expectations. I am delighted to raise my rating to 5. I am very grateful for having had the opportunity to read this excellent paper and to have deepened my understanding through this discussion.

Initially, I had doubts about whether the authors' music generation process truly incorporated auditory characteristics. However, the authors' responses completely alleviated my concerns (beyond my expectations). Therefore, I have updated my score to positive.

**Limitations:**

Yes. The authors clearly address the limitations.

As an additional request, if the authors could more clearly discuss the validity and limitations of the authors' use of outpainting and inpainting mechanisms to introduce psychophysiological auditory properties postprocessively instead of introducing them directly into the training of the generative AI, the authors would be happy to discuss the future of this paper. improvements to this paper in the future.

**Paper Formatting Concerns:**

I have not find any paper formatting concerns.

**Quality:**

2

**Strengths And Weaknesses:**

[Strengths]

- I believe that the focus proposed in this paper is highly insightful as one of the new applications of generative AI in recent years.

- The authors provide video materials as a tool to intuitively convey the effectiveness of the proposed method to readers. This is very useful for easily following the new problem setting assumed by the authors and understanding the practical effects of the proposed method.

- The use of subjective evaluation as an experiment to demonstrate the empirical effectiveness of the proposed method greatly enhances its persuasiveness.

[Weaknesses]

- There are concerns about whether the psychological auditory characteristics of humans are truly being utilized effectively in the generation of music for environmental noise (see question for details).

- I am concerned that the technology used to extract musical characteristics for generating music from noise (masking high-energy areas on the Mel spectrogram) is designed to fit the authors' application examples well.

---

> ### Author Rebuttal · Authors · 2025-07-30
>
> ## Rebuttal for reviewer cD4J
>
> We thank the reviewer for the thoughtful and encouraging feedback. We appreciate the recognition of our work as a *highly insightful* application of generative AI, as well as the positive remarks on the *clarity* of the writing, the *appeal* of the proposed direction, and the *empirical persuasiveness* of the subjective evaluation. We also value the acknowledgment of our system as a *practical and reasonable solution* to a socially relevant challenge. Below, we respond to the reviewer’s concerns in detail.
>
> >Q1: There are concerns about whether the psychological auditory characteristics of humans are truly being utilized effectively in the generation of music for environmental noise (see question for details). [Major concern 1: Validity of outpanting and inpainting mechanism instead of directly incorporating psyco-aural characteristics into generative AI] I am questioning whether it is possible to utilize the effects of acoustic masking based on psychological auditory characteristics at the stage of generating music.I understand that the effects of psychological auditory characteristics are utilized in Stage 2 of the authors' method. However, Stage 2 appears to retain only functions similar to an equalizer. To obtain generated music that utilizes psychological auditory characteristics as described in the introduction, it seems necessary to explicitly introduce these effects in Stage 1 of the authors' method.
>
> **A1:** Thank you for the thoughtful comment. We would like to clarify that the principle of auditory masking is considered **throughout both stages** of our framework, not just in Stage 2.
>
> 1. In Stage 1, we explicitly design the conditioning mask *M* based on the high-energy regions of the mel-spectrogram of the noise input. This choice is motivated by the established psychoacoustic theory, that frequencies with **higher local energy** are more perceptually salient and more likely to interfere with auditory perception [1,2].  Since the mel-spectrogram reflects energy distributions in critical bands, extracting the most energetic regions helps us locate the parts of noise that are most audible and thus most important to mask. By using these high-energy regions as input conditions, we guide the model to generate music that naturally aligns with the most perceptually prominent components of the noise, enabling more effective masking of those elements during blending.
> 2. Furthermore, to assess the sensitivity of the output to the mask design, we conducted additional experiments with alternative masking strategies:
>
> - Randomly selected 20% region
> - Larger high-energy area (30%–50%)
> - Our original 10%–25% high-energy mask
>
>     The results are summarized below:
>
>     | Masking Strategy   | FAD ↓  | KL ↓   |
>     |--------------------|--------|--------|
>     | Random 20% region | 17.82   | 2.47   |
>     | Top 30%–50% high-energy area     | 13.79   | 2.14   |
>     | Top 10%–25% high-energy (ours) | 7.98   | 1.67   |
>
>     As shown, altering the masking strategy significantly degrades performance. This underscores the critical role that **psychoacoustically informed mask selection** plays in the effectiveness of the first-stage generation.
> 3. In Stage 2, our adaptive amplification strategy builds directly upon the foundation laid in Stage 1. By conditioning on the high-energy regions of the noise during generation, the resulting music naturally inherits similar frequency distribution patterns in those regions. This design ensures that the musical signal already contains energy concentrated in the same areas as the noise. As a result, during Stage 2, we can achieve effective auditory masking by applying only modest amplification, since the generated music is **already aligned both temporally and spectrally with the dominant noise components**. This tightly coupled, two-stage design allows our system to perform perceptually effective blending with lower gain, preserving musical coherence while suppressing unwanted noise in a psychologically informed manner.
> 4. We agree that incorporating psychoacoustic knowledge directly into the generative model's learning objective is a promising direction and appreciate the reviewer’s encouragement in that regard; we consider it a compelling avenue for future research.
>
> >Q2: I am concerned that the technology used to extract musical characteristics for generating music from noise (masking high-energy areas on the mel-spectrogram) is designed to fit the authors' application examples well. [Major concern 2: Validity of prompt design (Mask M)] I do not fully understand the design guidelines for mask M, which is used to extract high-energy portions from noise. For example, if mask M is intended to extract rhythmic elements from noise, it seems possible to use a method that extracts periodic elements. Alternatively, if the goal is to extract frequency patterns that co-occur simultaneously from noise, it seems more appropriate to use a method that extracts low-rank structures from matrices, such as non-negative matrix factorization.I am unclear about the motivation behind the authors' decision to assign mask M the role of focusing solely on high-energy regions.
>
> **A2:** We appreciate the reviewer’s insightful suggestions regarding alternative ways of designing the conditioning mask 𝑀. We chose the current design based on the following considerations:
>
> 1. As mentioned in **A1**, our core goal is to **mask the most perceptually intrusive parts** of the environmental noise. According to psychoacoustic background, among all sounds perceptible on the loudness curve, the strongest (highest-energy) frequency components are the most prominent [1,2], thus, the most critical to address. By selecting the top 10–25% highest-energy regions in the mel-spectrogram as our mask 𝑀, we directly target the parts of the noise that listeners are most likely to notice and therefore most benefit from masking.
>
> 2. This masking-oriented design aligns well with our objective: we are not aiming to extract rhythmic or harmonic structure itself, but to generate a musically coherent signal that perceptually suppresses noise. While techniques such as periodicity analysis or matrix decomposition (e.g., NMF) are useful for uncovering latent structures, they *may not focus on the most perceptually relevant aspects* of the noise for our purpose.
>
> 3. Furthermore, this energy-based masking scheme naturally aligns with our Stage 2 amplification strategy: the same mask not only guides the music generation in Stage 1, but also **informs gain adjustment** in Stage 2. Because the generated music has already been conditioned to match the dominant energy regions of the noise, it becomes inherently more effective at masking those components. As a result, only a relatively small gain adjustment is needed to achieve perceptually meaningful suppression of the noise, thereby reducing the listener’s awareness of disruptive elements while preserving musical naturalness.
>
> 4. In summary, our design of mask *M* is not arbitrarily tailored to our examples but is instead a principled choice for maximizing masking effectiveness in a training-free setting. The high-energy strategy ensures that generation is perceptually aligned with the noise in a way that directly supports our task objective.
>
> We are glad to integrate the above clarifications into our revised manuscript.
>
> > As an additional request, if the authors could more clearly discuss the validity and limitations of the authors' use of outpainting and inpainting mechanisms to introduce psychophysiological auditory properties postprocessively instead of introducing them directly into the training of the generative AI.
>
> **A3:** Thank you for the valuable comment. We provided the extended discussion below:
>
> - **Validity of outpainting and inpainting mechanisms:** As discussed in **A1** and **A2**, our proposed inpainting and outpainting operations introduce psychophysiological auditory properties via masking strategies. These are applied at the spectrogram level and are designed to align with perceptual cues of salience and continuity.
>
> - **Advantages and limitations of post-sampling techniques versus training-time integration:**
>
>    - Our post-sampling strategy offers strong flexibility and generalization, as it is decoupled from the training data and model. This allows it to be applied across different generative backbones and to a wide range of real-world noise types. Our extended generalization results (please also see **A3** and **A4** to *R-nRgr*) empirically support this advantage.
>
>    - The main trade-off, however, is additional computational overhead during inference, due to the need to extract masking cues from the noise input. However, this overhead can be mitigated with modern fast-inference techniques, such as [3, 4].
>
>    - In contrast, integrating such mechanisms during training would require retraining generative models from scratch, an effort that is both computationally expensive and data-sensitive. Such training-based models often struggle with out-of-distribution (OOD) noise inputs, unfamiliar noise types may lead to incoherent outputs. Moreover, training on **large-scale music data** may raise copyright concerns, whereas our pipeline uses open-access, non-sensitive environmental noise data, avoiding such complications.
>
> We hope to integrate the above discussions into our revised manuscript if the reviewer find it helpful.
>
> ---
> [1] Hugo Fastl and Eberhard Zwicker. Psychoacoustics: Facts and Models. Springer-Verlag, Berlin, Heidelberg. 2006.
>
> [2] Moore, B. C. J.. An Introduction to the Psychology of Hearing (6th ed.). Brill. 2012
>
> [3] SoundCTM: Unifying Score-based and Consistency Models for Full-band Text-to-Sound Generation. ICLR25.
>
> [4] Presto! Distilling Steps and Layers for Accelerating Music Generation. ICLR25.

---

> ### Comment · Reviewer_cD4J · 2025-08-01
>
> I am very grateful to the authors for their thoughtful and enthusiastic responses. Almost all of my concerns (with one exception) have been addressed in a way that exceeds my initial expectations. Therefore, I would like to raise my rating by one point (I was originally on the negative side, but at this stage, my opinion has shifted significantly toward the positive). I now believe that this paper is ready to be discussed with various researchers at this kind of important conferences on machine learning and AI.
>
> However, I would also like to explain why I am not giving a higher rating. I am not fully convinced conceptually by the following points. Why is the music generated to harmonize with the high-energy noise regions extracted in Stage 1 (i.e., most perceptually prominent components of the noise) able to suppress noise? Wouldn't it instead emphasize the noise?
> For example, consider replacing the high-energy regions with "vocals". Wouldn't the accompaniment that harmonizes with the vocals end up emphasizing the vocals?
> Of course, from the experiments the authors have kindly added, I understand that the current system and scenario do not exhibit such undesirable behavior. However, personally, I still do not fully grasp why the system exhibits such desirable functionality. This is why I am unable to give a higher rating.
>
> I once again appreciate the authors' thoughtful responses.
>
> P.S., Sorry, it seems that the rating change will be made along with my (each reviewer's) final judgment, so I will make sure to address it at that time.

---

> > ### Author Response · Authors · 2025-08-02
> >
> > We sincerely thank the reviewer for the prompt and thoughtful reply. We are very glad that our previous responses have addressed most of your concerns. At the same time, we appreciate the insightful remaining conceptual question and the vocal-based counterexample raised. Below, we provide further clarification and discussion from three perspectives: (1) relevant insights from psychoacoustics, (2) particularly a **technical clarification** of our design intent, and (3) additional empirical results specifically addressing the vocal scenario you proposed.
> >
> > > Why is the music generated to harmonize with the high-energy noise regions extracted in Stage 1 able to suppress noise instead of emphasizing the noise?
> >
> > **1. Perceptual masking vs. physical suppression.**
> >
> > Our system does not physically suppress or remove the noise, but instead introduces a **more dominant auditory signal** to render the noise less perceptible to the human auditory system. This masking signal, shaped by the user-defined text prompt, is typically **musical, structured, and non-intrusive,** providing a more pleasant listening experience.
> >
> > This design leverages a well-established psychoacoustic principle: when two sounds overlap in time and frequency, the one with significantly higher energy masks the weaker one, making it inaudible even though it remains physically present. This phenomenon arises from the auditory system’s organization into overlapping frequency-selective filters known as critical bands [1]. Within a given band, only the dominant signal drives neural excitation beyond the perceptual threshold. As summarized by Moore (2012): “The tone that is suppressed (maskee) ceases to contribute to the pattern of phase locking, and the neuron responds as if only the suppressing tone (masker) was present.”
> >
> > This also explains why our system targets the **high-energy regions** of the noise. These components are the most perceptually prominent and the most difficult to blend naturally. If they can be effectively masked, the remaining low-energy components—already less intrusive—have minimal impact on the overall listening experience. Thus, our design assumes that masking the most salient regions is sufficient to suppress the overall perceptual presence of the noise.
> >
> > **2. Technical insight: harmonized generation via learned priors.**
> >
> > From a technical standpoint, our use of **inpainting and outpainting** in Stage 1 draws inspiration from similar techniques in computer vision [2,3], where a trained generative model can plausibly reconstruct missing regions based on partial input, e.g., generating a full body from a cropped face. In our context, we apply this principle in the audio domain: by conditioning on the most perceptually salient (high-energy) regions of the noise and a user-defined prompt, the model completes the rest of the music in a way that aligns both **semantically (prompt-guided)** and **structurally (spectro-temporally coherent)** with the input.
> >
> > This strategy ensures that the resulting musical component not only overlaps with the noise in time and frequency but also forms a **harmonious and plausible continuation**, thanks to the strong **prior learned by the generative model**. As a result, the generated music is more likely to mask the salient noise effectively while maintaining perceptual coherence.
> >
> > Notably, this technical design **applies to partial noise regardless of its energy levels**. In other words, **the learned prior from base music generative models ensures that our Stage 1 outputs a musical piece rather than an emphasized and repetitive version of the noise input**.
> >
> > **3. Empirical evidence on vocal masking.**
> >
> > We find the vocal case particularly interesting, as vocals differ from mechanical or environmental noise in spectral and temporal patterns. To explore this, we selected 100 vocal clips from the VocalSet dataset [4] as **maskees** and evaluated them using the same masking pipeline.
> > |Dataset|FAD↓|KL↓|
> > |-|-|-|
> > |EPIC-SOUNDS|7.98|1.67|
> > |ECS-50|6.76|2.14|
> > |VocalSet|7.39|1.86|
> >
> > The results on VocalSet are consistent with other noise types, indicating that our system maintains quality even for more structured and expressive inputs. This supports the generalizability of our masking-based strategy beyond simple background noise to more complex acoustic scenarios.
> >
> > We will include these additional results and discussion in the revised version of the manuscript. Please feel free to reach out with any further questions or suggestions. We sincerely appreciate your thoughtful engagement.
> >
> > ---
> >
> > [1] Moore, B. C. J.. An Introduction to the Psychology of Hearing (6th ed.). Brill. 2012
> > [2] Inpainting using Denoising Diffusion Probabilistic Models. CVPR2022.
> > [3] Continuous-Multiple Image Outpainting in One-Step via Positional Query and A Diffusion-based Approach. ICLR 2024.
> > [4] Wilkins, J., Prem Seetharaman, Alison Wahl, & Bryan Pardo. (2018). VocalSet: A Singing Voice Dataset (1.0) [Data set]. Zenodo.

---

> > > ### Comment · Reviewer_cD4J · 2025-08-03
> > >
> > > I am grateful for the authors' strong response and am both surprised and impressed. This is a very strong rebuttal. My concerns have been completely resolved beyond my expectations. I am delighted to raise my rating to 5. I am very grateful for having had the opportunity to read this excellent paper and to have deepened my understanding through this discussion.

---

> > > > ### Author Response · Authors · 2025-08-04
> > > >
> > > > We thank the reviewer for the approval of our work. We also appreciate the opportunity to benefit from this constructive exchange.

---

### Official Review · Reviewer_nRgr · 2025-07-05

**Clarity:** 3
**Significance:** 3
**Originality:** 3
**Rating:** 4
**Confidence:** 1

**Summary:**

This paper presents BNMusic, a diffusion-based model for music generation that blends environmental sounds (ENS) and musical notes. Unlike prior works that treat non-musical noise as background, BNMusic integrates ENS as a controllable and creative component of music. The method introduces a two-stage pipeline: first generating ENS audio, then conditioning music generation on it via diffusion. The approach is evaluated on a new dataset combining FSD50K and MusicCaps and demonstrates superior controllability and realism over baselines.

**Questions:**

- How would BNMusic perform on genres where environmental sounds are less common (e.g., classical)?
- Can the model adapt to real-time or interactive settings where ENS evolves dynamically?
- Is there a way to control the temporal alignment between ENS events and musical phrases?
- Would joint training (instead of two-stage) affect quality or controllability?
- Could more baselines (e.g., AudioLDM with explicit conditioning) be included?

**Ethical Concerns:**

["NO or VERY MINOR ethics concerns only", "Major Concern: Improper research involving human subjects"]

**Limitations:**

Yes

**Quality:**

3

**Strengths And Weaknesses:**

# Strengths:
- Novel framing of environmental sounds as a generative music component rather than noise.
- Effective use of a two-stage diffusion process for better disentanglement and controllability.
- High-quality samples and a new dataset (BNMusic-50K) contribute to reproducibility.
- Both objective (FC, FAD) and subjective (MOS) metrics show strong performance.

# Weaknesses:
- Limited baseline comparison; focuses on Mubert and simple concatenation.
- Lack of detailed ablation on the contribution of ENS vs music branch in training.
- Generalization beyond the proposed dataset is not fully explored.

---

> ### Author Rebuttal · Authors · 2025-07-30
>
> ## Rebuttal for reviewer nRgr
> We appreciate the valuable feedback from the reviewer and the recognition of our *novel framing of environmental sounds as a creative component*, *the effectiveness of our two-stage diffusion process*, and *the high-quality samples produced by BNMusic*.
>
> >Q1: Limited baseline comparison; focuses on Mubert and simple concatenation. Could more baselines (e.g., AudioLDM with explicit conditioning) be included?
>
> **A1:**
> Thank you for the suggestion. Following your advice, we additionally tested AudioLDM with explicit noise conditioning as a baseline. When using AudioLDM with noise as the direct audio condition, we observe that the model tends to incorporate excessive elements from the noisy input. This is likely because the audio condition provides a stronger generative prior than the accompanying text prompt, leading the model to focus more on reproducing the structure of the noise rather than generating harmonically coherent music. As a result, the output often resembles a more tonal version of the noise rather than distinct musical content.
> | Methods         |FAD ↓    | KL ↓   |
> |-----------------|---------|--------|
> | AudioLDM        | 27.06   | 3.80   |
> | BNMusic (Ours)  | 7.98    | 1.67   |
>
> In contrast, our method effectively separates the guidance signal from the raw noise using a two-stage design, which enables more perceptually coherent and better-integrated outputs. These additional baseline results will be included in the revised version to support our design choices.
>
> >Q2: Lack of detailed ablation on the contribution of ENS vs music branch in training. Would joint training (instead of two-stage) affect quality or controllability?
>
> **A2:**
> 1. Thank you for the question. We would like to firstly clarify that our method does **not** involve any additional or joint training. Instead, we focus on **re-designing the inference pipeline** in Stage 1 and applying an **adaptive amplification strategy** in Stage 2. Both stages are **training-free** and built to fully utilize the capabilities of the **pre-trained base models**.
> 2. As shown in Tab. 3 (p.9) of our original manuscript (also included below for reference), we present an ablation study that evaluates the contributions of each component in our proposed pipeline. The results show that both the **Stage 1 generation** and the **Stage 2 amplification** play crucial roles in improving the perceptual and quantitative quality of the output. This demonstrates that each part of our design contributes meaningfully to the final result.
>     | Outpainting (Stage 1)| Inpainting (Stage 1) | Adaptive Amplification (Stage 2)         |FAD ↓    | KL ↓   |
>     |-----------------|---------|--------|--------|--------|
>     |     ✘     |      ✘      |    ✘    | 34.17   | -   |
>     |     ✓     |      ✘      |    ✘    | 8.68   | 1.89   |
>     |     ✓     |      ✘      |    ✓    | 9.18   | 1.84   |
>     |     ✓     |      ✓      |    ✘    | 8.00   | 1.78   |
>     |     ✓     |      ✓      |    ✓    | 7.98   | 1.67   |
>
> 3. Given the nature of our task, we believe the quality and controllability are closely intertwined in our setting. Our objective is to use the environmental noise as a condition to guide music generation, such that the generated music perceptually integrates with the noise and reduces the listener’s awareness of it. In this context, effective controllability means that the noise appropriately influences the music generation, such as alignment, while the music remains sufficiently harmonious and pleasant. Achieving this balance is precisely what ensures the quality of the final blended output.
>
> >Q3: Generalization beyond the proposed dataset is not fully explored.
>
> **A3:** Thank you for raising this point. As described in Section 4.2 of the original manuscript, we evaluated our method on two distinct datasets to assess its generalization ability. We used noise samples from the **EPIC-SOUNDS** dataset, which encompass 58 distinct action types and 140 different object types. We also included the **ESC-50** dataset, which contains 50 different categories of environmental sounds. The strong and consistent performance of our method on both datasets demonstrates its robustness across a wide range of realistic auditory conditions.
>
> To further address the reviewer’s concern about generalization, we collected 10 types of everyday environmental noises (elevator operation sounds, air conditioner hums, keyboard typing, etc) and used them as new test cases. The results are shown below:
> |Noises|Recorded noise (FAD ↓)|EPIC-SOUNDS (FAD ↓)|ESC-50 (FAD ↓)|
> |-|-|-|-|
> |Real Music|9.56|15.41|8.32|
> |Riff-A2A |8.75|10.95|7.74|
> |MusicGen|9.08|13.15|9.11|
> |BNMusic (Ours)|7.53|7.98|6.76|
>
> These results further validate the practical generalization ability of our framework. Moreover, since our method is entirely training-free and does not involve any fine-tuning or learning from the input data, it is inherently less susceptible to dataset-specific bias. This design allows our approach to generalize effectively to unseen acoustic conditions without any adaptation or retraining.
>
> >Q4: How would BNMusic perform on genres where environmental sounds are less common (e.g., classical)?
>
> **A4:**
> 1. As discussed in **A3**, our method demonstrates strong generalization capabilities and is able to handle a wide variety of environmental sounds, regardless of whether they are commonly seen or relatively rare in real-world settings.
> 2. To further address the reviewer’s concern, we conducted additional experiments focusing specifically on less common types of environmental sounds. We selected two subsets: the 30 least frequent categories and the 50 least frequent categories from our test pool. The results are shown below:
>     |Methods|50 least frequent ENS (FAD ↓)|All ENS (FAD ↓)|
>     |-|-|-|
>     |Riff-A2A|13.08|13.15|
>     |MusicGen|11.20|10.95|
>     |BNMusic (Ours)|8.38|7.98|
>
>     These results suggest that even when applied to rarer or less familiar noise types, BNMusic maintains strong performance.
>
> 3. Regarding the “*Classical*” genre mentioned in your question, we would like to clarify that in our framework, this refers to the musical style specified by the user prompt, rather than a characteristic of the environmental noise itself. To examine genre sensitivity, we evaluated different music styles,
>
>     |Genres|Classical (FAD ↓)  |Jazz (FAD ↓) |Pop (FAD ↓) |Rock (FAD ↓) |
>     |-|-|-|-|-|
>     |BNMusic (Ours) |10.53|9.51|7.99|7.89|
>
>     The results suggest that genres like *Classical* and *Jazz*, which tend to have smoother dynamics and slower tempos, may be less effective at perceptually masking noise due to their limited rhythmic overlap with common environmental sounds. In contrast, more upbeat genres like *Pop* and *Rock* tend to blend more naturally with a wider range of noise patterns, making them more robust across diverse scenarios.
>
> >Q5: Can the model adapt to real-time or interactive settings where ENS evolves dynamically?
>
> **A5:**
> 1. Our method is particularly well suited for many real-world scenarios where environmental noise exhibits temporal consistency or repetition, such as elevator operation, automatic garage doors, or air conditioning systems. In addition, there are scenarios like subway cars, where the sound patterns during uniform motion are highly similar across different cars. This makes it feasible to use the noise recorded in the front car as input to generate blending music that can be played in another car, achieving effective noise masking without requiring strictly real-time processing.
> 2. Moreover, the most time-consuming stage of our method—music generation—can be significantly accelerated using recent fast diffusion inference techniques, such as SoundCTM [1] and Presto [2]. Based on the speed-up ratios reported in these works, a 5-second audio clip can be generated in under 0.1 seconds with single GPU inference. Combined with our preprocessing and Stage 2 adaptive amplification (which take approximately 0.3 seconds in total), the full pipeline can operate within 0.5 seconds. With further advances in hardware and model optimization, even faster performance can be expected, making our method promising for near-real-time or interactive applications where noise evolves gradually rather than abruptly.
>
> >Q6: Is there a way to control the temporal alignment between ENS events and musical phrases?
>
> **A6:** Thank you for the question. In our method, the mel-spectrogram serves as a time-frequency representation, where the horizontal axis corresponds to time and the vertical axis to frequency, with the intensity at each point reflecting the energy at a given time and frequency. When we extract the high-energy region of the ENS spectrogram to serve as the condition for outpainting, we are effectively providing the model with both spectral and temporal cues. As a result, the generative process naturally preserves the temporal structure encoded in the input, allowing the generated musical content to align with salient ENS events over time. This implicit alignment enables the music to respond rhythmically and structurally to the dynamics of the original noise. As shown in Figure 7 in the Appendix (Page 15) of the original manuscript, when the ENS contains salient events—such as sudden onsets or accents—the model tends to generate corresponding musical peaks at aligned time positions, demonstrating strong temporal synchronization. This inherent matching ability of the outpainting-inpainting pipeline distinguishes our approach from other models like MusicGen, which, despite conditioning on noise, do not achieve such precise temporal alignment, thus highlighting the advantage of our method in producing time-aware and contextually coherent musical outputs.
>
> ---
>
> [1] SoundCTM: Unifying Score-based and Consistency Models for Full-band Text-to-Sound Generation. ICLR25.
>
> [2] Presto! Distilling Steps and Layers for Accelerating Music Generation. ICLR25.

---

### Decision · Program_Chairs · 2025-09-17

**Decision:**

Accept (poster)

**Comment:**

The paper proposes BNMusic, a novel method for blending environmental noise into personalized music using a two-stage generative framework. It creatively reframes noise as input rather than interference.

Strengths:
- The idea of treating environmental noise as a creative input for music generation is original and potentially impactful.
- Strong empirical results, including both objective metrics and human evaluations.
- The writing is clear and well-organized, making the methodology and contributions easy to understand.

Although the real-time applicability is still uncertain, I recommend acceptance as the paper presents a novel idea, supported by strong technical execution and thoughtful responses during the rebuttal.